# EFFICIENT MODEL EDITING WITH TASK-LOCALIZED SPARSE FINE-TUNING

**Leonardo Iurada**[1,*]**, Marco Ciccone**[2]**, Tatiana Tommasi**[1]
[1]Politecnico di Torino, Italy     [2]Vector Institute, Toronto, Ontario, Canada
[*]Correspondance to: `leonardo.iurada@polito.it`

## ABSTRACT

Task arithmetic has emerged as a promising approach for editing models by representing task-specific knowledge as composable task vectors. However, existing methods rely on network linearization to derive task vectors, leading to computational bottlenecks during training and inference. Moreover, linearization alone does not ensure weight disentanglement, the key property that enables conflict-free composition of task vectors. To address this, we propose `TaLoS` which allows to build sparse task vectors with minimal interference without requiring explicit linearization and sharing information across tasks. We find that pre-trained models contain a subset of parameters with consistently low gradient sensitivity across tasks, and that sparsely updating only these parameters allows for promoting weight disentanglement during fine-tuning. Our experiments prove that `TaLoS` improves training and inference efficiency while outperforming current methods in task addition and negation. By enabling modular parameter editing, our approach fosters practical deployment of adaptable foundation models in real-world applications[1].

## 1   INTRODUCTION

Large pre-trained models (Radford et al., 2021; Raffel et al., 2020; Brown et al., 2020) have become the cornerstone of modern machine learning, showcasing impressive capabilities across a broad spectrum of tasks. Currently, their development is confined to a few computationally and financially well-resourced research groups, but once publicly released they provide a wealth of reusable knowledge that greatly benefits downstream applications. Indeed, fine-tuning large models to achieve optimal performance on specialized tasks or to align with user preferences is becoming an increasingly democratized practice, thanks to efficient methods enabling model customization on affordable consumer GPUs. Parameter-efficient fine-tuning (PEFT) (Hu et al., 2022; Liu et al., 2022; 2024), sparsity (Ansell et al., 2022; 2024), and quantization (Dettmers et al., 2023) are some of the techniques that fueled the growth of a rich ecosystem of task-specific models. They are, in turn, readily shared on open platforms (Pfeiffer et al., 2020; Poth et al., 2023) fostering collaborative knowledge building by enabling users to adapt and integrate specialized modules (Raffel, 2023).

In this context, *task arithmetic* (Ilharco et al., 2023) has emerged as a promising framework for scalable and cost-effective model editing. It encodes task-specific knowledge using *task vectors*, derived by fine-tuning a pre-trained model and subtracting its original weights from the fine-tuned ones. Task vectors can be combined through addition and subtraction to enhance specific tasks, suppress undesired behaviors, or merge functionalities. However, when task vectors are independently fine-tuned in decentralized collaborative settings, task interference becomes a significant concern (Yadav et al., 2023; Wang et al., 2024), as adding or removing a functionality disrupts previously acquired knowledge. Task interference occurs when fine-tuning modifies parameters that are critical to other tasks, resulting in unintended behavioral shifts. To prevent this, data from disjoint regions in the input space (representing different tasks) should affect only their corresponding regions in the activation space. Ortiz-Jimenez et al. (2023) formalized this concept as *weight disentanglement*. Their research showed that this property is an emergent feature of pre-training, which makes foundation models inherently suited for task arithmetic. The key question therefore becomes: *how can fine-tuning preserve weight disentanglement?*

Explicitly linearizing the model during fine-tuning has been a promising direction to maintain weight disentanglement, albeit with increased computational overhead (Ortiz-Jimenez et al., 2023). In this

---

[1]Code available at: `https://github.com/iurada/talos-task-arithmetic`

work, we first show that model linearization alone is not sufficient, as its task functions can still activate for arbitrary inputs. Instead, we propose a set of *function localization* constraints to exactly implement the weight disentanglement property on linearized networks. Then, we introduce a novel *sparse fine-tuning* approach that implements such constraints while avoiding the need for explicit model linearization. The proposed method strategically updates a subset of model parameters, simultaneously promoting linearized behavior and enforcing function localization. Extensive empirical analyses and theoretical justifications demonstrate that our approach *effectively promotes weight disentanglement*, ensuring compatibility between task vectors without the need for sharing information between users and tasks. This enables efficient and robust model editing through the simple addition and subtraction of sparse task vectors, facilitating decentralized collaborative strategies.

**We can summarize our main contributions as follows.**

- We advance the field of task arithmetic by deriving a novel set of function localization constraints that provide exact guarantees of weight disentanglement on linearized networks.

- We empirically observed that the least sensitive parameters in transformer-based architectures pre-trained on large-scale datasets can be consistently identified regardless of the task. We exploit this regularity to satisfy the localization constraints under strict individual training assumptions.

- We introduce *Task-Localized Sparse Fine-Tuning* (`TaLoS`) that enables task arithmetic by jointly implementing the localization constraints and inducing a linear regime during fine-tuning, without incurring in the overheads of explicit network linearization.

Overall, our work addresses a critical gap in task arithmetic, providing a more complete and practical framework for parameter-space model editing, targeting real-world applications.

## 2 RELATED WORKS

**Sparsity & Parameter-Efficient Fine-Tuning.** Sparsity has emerged as a fundamental concept in efficient deep learning, manifesting in both training and adaptation methodologies. Sparse fine-tuning strategies (Guo et al., 2021; Xu et al., 2021) improve training efficiency by selectively updating subsets of model parameters. These approaches often leverage the Fisher information matrix (Fisher, 1922; Amari, 1996) to identify important weights for updating (Sung et al., 2021; Ben Zaken et al., 2022) or, conversely, focus on fine-tuning only the least important parameters to minimize disruption of the original model's knowledge (Liao et al., 2023; Ansell et al., 2024). Sparse masking techniques (Wortsman et al., 2020; Mallya et al., 2018; Mallya & Lazebnik, 2018; Havasi et al., 2020) further exploit this principle by employing subnetworks for continual and multi-task learning. Parameter-efficient fine-tuning (PEFT) represents another approach to adaptation with minimal parameter updates. Popular PEFT methods include adapter layers (Houlsby et al., 2019), prefix tuning (Li & Liang, 2021), and low-rank adaptation (LoRA, (Hu et al., 2022)). LoRA in particular approximates model updates through rank decomposition matrices while keeping pre-trained weights frozen. In a complementary direction, Ansell et al. (2022); Panda et al. (2024) investigate sparse weight addition as a flexible approach to model composition. These sparse adaptation techniques connect to the broader field of model pruning, which has traditionally been applied post-training for efficient storage and inference (Blalock et al., 2020). The Lottery Tickets Hypothesis (Frankle & Carbin, 2019) expanded this idea by demonstrating that sparse subnetworks identified at initialization can, when trained, match the performance of the original dense model while significantly reducing computational costs.

**Model Merging.** The goal of model merging is to combine multiple task-specific models into a single multi-task model without performing additional training. This requires merging techniques that prevent negative interferences among separately learned parameters. While simple parameter averaging can be effective, particularly when fine-tuned models share the same initialization (Wortsman et al., 2022; Ramé et al., 2023), it does not always yield optimal results. As a result, existing approaches explored tailored re-weighting schemes, though these often come with high computational costs. RegMean (Jin et al., 2023) solves a local linear regression problem for each individual linear layer in the model that requires transmitting extra data statistics of the same size as the model and additional inference steps. Fisher Merging (Matena & Raffel, 2022) exploits the Fisher Information Matrix. This method, however, requires computing gradients, resulting in high memory costs. A recent approach exploits extra unlabeled data to learn the model merging weights (Yang et al., 2024).

**Task Arithmetic.** Task arithmetic (Ilharco et al., 2023) was introduced as a paradigm for editing models based on arithmetic operations over *task vectors* obtained by fine-tuning a base pre-trained model and then subtracting the pre-trained weights from the fine-tuned ones. This concept has also been used in model merging, with methods that prepare task vectors before adding them together

to produce a single multi-task model. Recent examples of this strategy are TIES-Merging (Yadav et al., 2023) which resolves parameter overlap and sign conflicts after merging using heuristics, and TALL Masks / Consensus (Wang et al., 2024) that deactivates irrelevant parameters through binary masking. Other approaches sparsify task vectors by randomly dropping and rescaling parameters (Yu et al., 2024) or masking weight outliers (Davari & Belilovsky, 2024). However, task arithmetic goes beyond model merging as it aims at *adding to* or *deleting* knowledge and capabilities *from* a model in a modular and efficient manner. Its effectiveness relies on weight disentanglement, a property emerging during pre-training, as shown by Ortiz-Jimenez et al. (2023). They proposed to preserve weight disentanglement by fine-tuning in the tangent space via full model linearization with high computational costs. To improve efficiency, Tang et al. (2024) proposed to use linearized low-rank adapters in the attention modules during fine-tuning. Still, linearization alone does not guarantee task localization, potentially letting weight disentanglement decrease during fine-tuning.

Our work fits within task arithmetic as a PEFT approach to construct *sparse task vectors*. By leveraging strategies from pruning and sparse fine-tuning, we introduce a parameter update criterion that induces a linearized regime without explicit linearization and ensures functional task localization.

## 3 BACKGROUND

Consider a neural network $f$ with parameters $\boldsymbol{\theta} \in \mathbb{R}^m$, pre-trained on a mixture of tasks $\mathcal{P}$ to obtain parameters $\boldsymbol{\theta}_0$. We are interested in fine-tuning the pre-trained model $f(\cdot, \boldsymbol{\theta}_0)$ on a set of $T$ distinct classification tasks, with associated non-intersecting task data support $\mathcal{D} = \{\bigcup_{t=1}^T \mathcal{D}_t\} \subseteq \mathcal{D}_{\mathcal{P}}$ (*i.e.* $\forall t, t'$ if $t \neq t'$ then $\mathcal{D}_t \cap \mathcal{D}_{t'} = \varnothing$).

In this setting, the core idea behind task arithmetic, introduced in Ilharco et al. (2023), is to represent the knowledge acquired for each task $t$ as a *task vector* $\boldsymbol{\tau}_t = \boldsymbol{\theta}_t^\star - \boldsymbol{\theta}_0$, obtained by subtracting the initial parameters from the fine-tuned parameters. Intuitively, this vector captures the direction and magnitude of change in the model's weight space induced by learning task $t$. By manipulating tasks via task arithmetic operations we can effectively add, combine, or remove knowledge in the pre-trained model producing actual functional behaviors directly in the parameters space.

As formalized by Ortiz-Jimenez et al. (2023), a network $f$ is said to satisfy the task arithmetic property around $\boldsymbol{\theta}_0$ if it holds

$$f\left(\boldsymbol{x}, \boldsymbol{\theta}_0 + \sum_{t=1}^T \alpha_t \boldsymbol{\tau}_t\right) = \begin{cases} f(\boldsymbol{x}, \boldsymbol{\theta}_0 + \alpha_t \boldsymbol{\tau}_t) & \boldsymbol{x} \in \mathcal{D}_t \\ f(\boldsymbol{x}, \boldsymbol{\theta}_0) & \boldsymbol{x} \notin \bigcup_{t=1}^T \mathcal{D}_t \end{cases} \tag{1}$$

with scaling factors $(\alpha_1, ..., \alpha_T) \in \mathcal{A} \subseteq \mathbb{R}^T$. This equation essentially states that adding a linear combination of task vectors to the initial parameters $\boldsymbol{\theta}_0$ is equivalent to selectively applying each task-specific modification to the model. In other words, the performance of the pre-trained model on different tasks can be modified independently if the task vector $\boldsymbol{\tau}_t$ does not modify the output of the model outside $\mathcal{D}_t$.

To fulfill the task arithmetic property, Ortiz-Jimenez et al. (2023) states that the model $f$ must exhibit a form of *weight disentanglement* with respect to the set of fine-tuning tasks, *i.e.*, $f$ should behave as a composition of spatially localized components corresponding to functions that vanish outside the task's data support. In practice, Equation 1 can be re-written as

$$f\left(\boldsymbol{x}, \boldsymbol{\theta}_0 + \sum_{t=1}^T \alpha_t \boldsymbol{\tau}_t\right) = f(\boldsymbol{x}, \boldsymbol{\theta}_0)\mathbb{1}\left(\boldsymbol{x} \notin \bigcup_{t=1}^T \mathcal{D}_t\right) + \sum_{t=1}^T f(\boldsymbol{x}, \boldsymbol{\theta}_0 + \alpha_t \boldsymbol{\tau}_t)\mathbb{1}(\boldsymbol{x} \in \mathcal{D}_t) \tag{2}$$

$$= g_0(\boldsymbol{x}) + \sum_{t=1}^T g_t(\boldsymbol{x}, \alpha_t \boldsymbol{\tau}_t) . \tag{3}$$

where $g_t(\boldsymbol{x}, \alpha_t \boldsymbol{\tau}_t) = \boldsymbol{0}$ for $\boldsymbol{x} \notin \mathcal{D}_t$ and $t = 1, ..., T$, and $g_0(\boldsymbol{x}) = 0$ for $\boldsymbol{x} \in \bigcup_{t=1}^T \mathcal{D}_t$, capturing the base behavior of the pre-trained model on inputs outside any of the task support.

Previous works (Tang et al., 2024; Ortiz-Jimenez et al., 2023) have sought to achieve task arithmetic by focusing on linearized neural networks (Ortiz-Jiménez et al., 2021), as they explicitly constrain $f$ to be represented as a linear combination of functions. Specifically, the linearization of $f$ can be achieved by its first-order Taylor expansion centered around $\boldsymbol{\theta}_0$:

$$f(\boldsymbol{x}, \boldsymbol{\theta}_0 + \alpha_t \boldsymbol{\tau}_t) \approx f_{\text{lin}}(\boldsymbol{x}, \boldsymbol{\theta}_0 + \alpha_t \boldsymbol{\tau}_t) = f(\boldsymbol{x}, \boldsymbol{\theta}_0) + \alpha_t \boldsymbol{\tau}_t^\top \nabla_{\boldsymbol{\theta}} f(\boldsymbol{x}, \boldsymbol{\theta}_0) . \tag{4}$$

The model $f_{\text{lin}}(\boldsymbol{x}, \boldsymbol{\theta}_0 + \boldsymbol{\tau}_t)$ represents a linearized neural network. For this type of networks, when combining together multiple task vectors, it holds

$$f_{\text{lin}}\left(\boldsymbol{x}, \boldsymbol{\theta}_0 + \sum_{t=1}^{T} \alpha_t \boldsymbol{\tau}_t\right) = f(\boldsymbol{x}, \boldsymbol{\theta}_0) + \sum_{t=1}^{T} \alpha_t \boldsymbol{\tau}_t^\top \nabla_{\boldsymbol{\theta}} f(\boldsymbol{x}, \boldsymbol{\theta}_0) \, . \tag{5}$$

While Equation 5 appears to closely resemble the weight disentanglement condition presented in Equation 3, this similarity is superficial unless each term $\alpha_t \boldsymbol{\tau}_t^\top \nabla_{\boldsymbol{\theta}} f(\boldsymbol{x}, \boldsymbol{\theta}_0)$ corresponds to a function that vanishes outside its task data support (*i.e.* it is *localized* within $\mathcal{D}_t$). In the following, we will demonstrate how to efficiently impose a condition of function localization.

## 4 TASK-LOCALIZED SPARSE FINE-TUNING

To formalize the condition of function localization for task arithmetic, we begin by revisiting the linear approximation of $f$ used in linearized fine-tuning. For Equation 5 to satisfy the weight disentanglement conditions in Equation 3, we must ensure that each $t$-th task-specific function $\boldsymbol{\tau}_t^\top \nabla_{\boldsymbol{\theta}} f(\boldsymbol{x}, \boldsymbol{\theta}_0)$ is active (non-zero) only for inputs within its corresponding task support, *i.e.*, $\boldsymbol{x} \in \mathcal{D}_t$. This requirement can be expressed as a set of constraints:

$$\forall \boldsymbol{x} \in \mathcal{D}_{t' \neq t}, \quad \boldsymbol{\tau}_t^\top \nabla_{\boldsymbol{\theta}} f(\boldsymbol{x}, \boldsymbol{\theta}_0) = 0 \, . \tag{6}$$

Satisfying these conditions ensures that updating the model's weights by training on task $t$ does not affect how the model processes data from other tasks, preventing interference between task vectors.

Directly implementing Equation 6 poses a significant practical challenge. Enforcing the constraint $\forall \boldsymbol{x} \in \mathcal{D}_{t'}$ requires simultaneous access to data from all other tasks ($t' \neq t$) during fine-tuning on task $t$. However, this is an impractical requirement in realistic settings where contributors optimize their model asynchronously on private, task-specific data. To address this, we assume that during pre-training the model is exposed to a vast mixture of tasks, including some that are similar to the $T$ fine-tuning tasks under consideration. Consequently, we expect the gradients $\nabla_{\boldsymbol{\theta}} f(\cdot, \boldsymbol{\theta}_0)$ to exhibit a shared structure across tasks, thereby bypassing the need for accessing all task data during fine-tuning.

### 4.1 FUNCTION LOCALIZATION UNDER INDIVIDUAL TRAINING CONSTRAINTS

As the gradient $\nabla_{\boldsymbol{\theta}} f(\boldsymbol{x}, \boldsymbol{\theta}_0)$ quantifies the influence of each parameter on the model's output for a given input $\boldsymbol{x}$, it serves as a direct measure of *parameter sensitivity*, describing how small variations in each parameter affect the model's input-output behavior.

Consequently, to satisfy the function localization constraints in Equation 6, our goal is to identify those parameters that have minimal impact on the model. In particular, by denoting the $j$-th element of $\boldsymbol{\theta} \in \mathbb{R}^m$ as $\boldsymbol{\theta}_{[j]}$, we define the *least-sensitive* parameters as the ones for which $\nabla_{\boldsymbol{\theta}_{[j]}} f(\boldsymbol{x}, \boldsymbol{\theta}_0) \approx 0$. We hypothesize that such parameters remain *least sensitive* across all tasks (*i.e.* $\forall \boldsymbol{x} \in \mathcal{D}$) and can thus be determined independently of the specific task, without having to access all task data.

To test our hypothesis, we conduct a sensitivity analysis following Chaudhry et al. (2018); Pascanu & Bengio (2013); Matena & Raffel (2022). We define $f(\boldsymbol{x}, \boldsymbol{\theta}_0) \triangleq \log p_{\boldsymbol{\theta}_0}(y|\boldsymbol{x})$, where $p_{\boldsymbol{\theta}_0}(y|\boldsymbol{x})$ denotes the probability of assigning class $y$ to $\boldsymbol{x}$. To quantify how changes in the parameters influence the model's output, we rely on the Fisher Information matrix (FIM) (Fisher, 1922; Amari, 1996), a positive semi-definite symmetric matrix given by:

$$F(\boldsymbol{\theta}_0, \mathcal{D}_t) = \mathbb{E}_{\boldsymbol{x} \sim \mathcal{D}_t}[\mathbb{E}_{y \sim p_{\boldsymbol{\theta}_0}(y|\boldsymbol{x})}[\nabla_{\boldsymbol{\theta}} \log p_{\boldsymbol{\theta}_0}(y|\boldsymbol{x}) \nabla_{\boldsymbol{\theta}} \log p_{\boldsymbol{\theta}_0}(y|\boldsymbol{x})^\top]].$$

For a parameter with index $j \in 1, \ldots, m$, the corresponding value on the diagonal of the FIM represents its sensitivity,

$$F_{[j,j]}(\boldsymbol{\theta}_0, \mathcal{D}_t) = \frac{1}{N} \sum_{i=1}^{N} \mathbb{E}_{y \sim p_{\boldsymbol{\theta}_0}(y|\boldsymbol{x}_i)}[\nabla_{\boldsymbol{\theta}_{[j]}} \log p_{\boldsymbol{\theta}_0}(y|\boldsymbol{x}_i)]^2 \, , \tag{7}$$

where $\boldsymbol{x}_1, \ldots, \boldsymbol{x}_N \in \mathcal{D}_t$ are i.i.d. examples, while the expectation on the output can be computed via sampling from the distribution of $p_{\boldsymbol{\theta}_0}(y|\boldsymbol{x}_i)$. The lower $F_{[j,j]}(\boldsymbol{\theta}_0, \mathcal{D}_t)$, the less the model will be affected by the $j$-th parameter changes.

**Least sensitive parameters are shared across tasks.** To study the role of the least sensitive parameters across tasks, we performed a pruning experiment, illustrated in Figure 1. We first

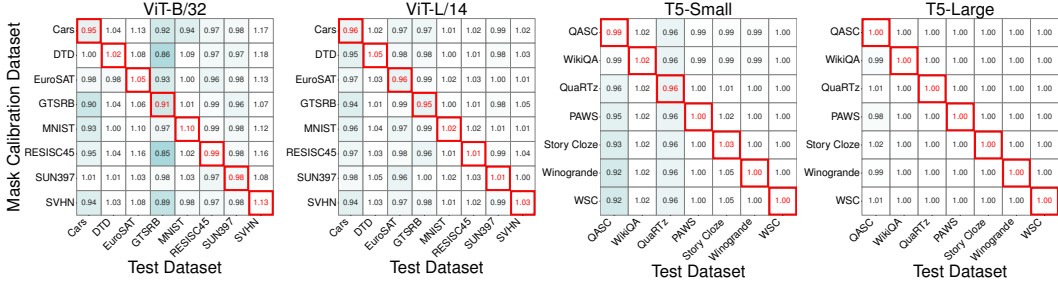

Figure 1: **Relative performance when pruning parameters with low sensitivity.** The heatmaps illustrate the effect of pruning the parameters with the lowest sensitivity (measured by $[F_{[j,j]}(\boldsymbol{\theta}_0, \mathcal{D}_t)]_{j=1}^m$) on different tasks across various pre-trained models using data from different tasks. Each grid compares the accuracy ratios for models after pruning, where the rows represent the task dataset $\mathcal{D}_t$ used to identify the parameters with the lowest sensitivity, and the columns show the model's zero-shot performance on each task after pruning those parameters. The accuracy ratios are normalized by the model's performance before pruning. The sparsity ratio (10%) was found as the maximal sparsity that minimally influenced the model's output on the mask calibration dataset.

identified the parameters with the lowest $F_{[j,j]}(\boldsymbol{\theta}_0, \mathcal{D}_t)$ by using only data from task $t$. We then pruned these parameters from the network and evaluated its performance on $t$ and other tasks $t' \neq t$. The results show that the pruned model retains its *zero-shot* performance over all tasks. We conclude that the least sensitive parameters can be effectively identified independently of the specific task, empirically supporting our hypothesis (further validation of this phenomenon and discussion in Appendix A.7).

Consequently, *function localization* can be achieved by updating only the least sensitive parameters, as for such updates the resulting dot product in Equation 6 is expected to be minimal across all tasks (we will expand on this in Section 4.3). Thus, we propose learning task vectors via a selective *Task-Localized Sparse Fine-Tuning* (TaLoS), wherein *only the parameters with the lowest sensitivity are sparsely updated during fine-tuning*.

## 4.2 TaLoS IMPLEMENTATION

Sparse fine-tuning consists in introducing a binary mask $\boldsymbol{c} \in \{0, 1\}^m$ to control which parameters are updated during gradient descent. Specifically, at each $i$-th iteration, the update rule becomes:

$$\boldsymbol{\theta}^{(i)} = \boldsymbol{\theta}^{(i-1)} - \gamma[\boldsymbol{c} \odot \nabla_{\boldsymbol{\theta}}\mathcal{L}(f(\boldsymbol{x}, \boldsymbol{\theta}^{(i-1)}), y)], \qquad (8)$$

where $\gamma$ is the learning rate, $\mathcal{L}$ is the loss function, and $\odot$ represents the element-wise product.

To achieve function localization we selectively update only the parameters with minimal impact on the model's output. Based on what was discussed earlier, we score each parameter using the diagonal elements of the FIM[2] $\boldsymbol{s} = [F_{[j,j]}(\boldsymbol{\theta}_0, \mathcal{D}_t)]_{j=1}^m \in \mathbb{R}^m$ and sort them to identify the index $j^*$ of the $k$-th lowest element in $\boldsymbol{s}$. This value is adopted as a threshold and we set $\boldsymbol{c}_{[j]} = 0$ if $\boldsymbol{s}_{[j]} \geq \boldsymbol{s}_{[j^*]}$ effectively freezing these parameters. Otherwise, $\boldsymbol{c}_{[j]} = 1$, allowing these parameters to be updated during fine-tuning. Note that the estimation of $\boldsymbol{c}$ may be susceptible to gradient noise (Tanaka et al., 2020). Thus, we follow standard Pruning-at-Initialization practices (Tanaka et al., 2020) and iteratively refine $\boldsymbol{c}$ in multiple rounds (we provide full details of TaLoS, alongside its pseudocode in Appendix A.2).

## 4.3 INSIGHTS ON SPARSITY AND LINEAR BEHAVIOR

**TaLoS promotes linear behavior.** Parameters with the smallest (ideally near-zero) $F_{[j,j]}(\boldsymbol{\theta}_0, \mathcal{D}_t)$ are associated with flatter regions in the loss landscape, as for $\boldsymbol{\theta}_0$ the FIM equals the Gauss-Newton approximation of the Hessian (Pennington & Worah, 2018; Kunstner et al., 2019). Updating parameters in a flat subspace allows the gradient to be approximately constant throughout fine-tuning, a necessary condition for operating in the linearized regime (Malladi et al., 2023b). This means that fine-tuning the least sensitive parameters *inherently promotes a linear behavior* without requiring explicitly linearizing the network. We follow Ortiz-Jimenez et al. (2023) to confirm this claim in Appendix A.4.

---

[2]Sensitivity scoring can be implemented through different approaches, as long as they preserve the same ranking as the FIM. For instance, given a scalar output and $f(\boldsymbol{x}, \boldsymbol{\theta}_0) \triangleq \log p_{\boldsymbol{\theta}_0}(y|\boldsymbol{x})$, $\mathbb{E}_{\boldsymbol{x}}[\|\nabla_{\boldsymbol{\theta}} f(\boldsymbol{x}, \boldsymbol{\theta}_0)\|]$ yields the same ranking as the diagonal of the FIM $\mathbb{E}_{\boldsymbol{x}}[[\nabla_{\boldsymbol{\theta}} \log p_{\boldsymbol{\theta}_0}(y|\boldsymbol{x}_i)]^2]$, as the absolute value function ($h(x) = |x|$) and the squaring function ($h(x) = x^2$) are both monotonically increasing in the interval $]0, +\infty[$.

**Function localization in `TaLoS`.** Given the *least sensitive* parameters are shared across tasks, the function localization constraints of Equation 6 for `TaLoS` can be rewritten and upper bounded as

$$\forall \boldsymbol{x} \in \mathcal{D}_t, \; |\boldsymbol{c} \odot (\boldsymbol{\tau}_t^\top \nabla_{\boldsymbol{\theta}} f(\boldsymbol{x}, \boldsymbol{\theta}_0)| \leq \|\boldsymbol{c} \odot \boldsymbol{\tau}_t\| \cdot \max_{\boldsymbol{x} \in \mathcal{D}_t} \|\boldsymbol{c} \odot \nabla_{\boldsymbol{\theta}} f(\boldsymbol{x}, \boldsymbol{\theta}_0)\| \leq k^2 \cdot \mu \cdot \eta \,. \quad (9)$$

Here, $\eta = \max_{\boldsymbol{x}} |\nabla_{\boldsymbol{\theta}_{[j^*]}} f(\boldsymbol{x}, \boldsymbol{\theta}_0)|$ is the magnitude of the $k$-th largest gradient element, capturing the maximum sensitivity of the fine-tuned parameters to input data. $\mu = \max_j |\boldsymbol{c}_{[j]} \odot \boldsymbol{\tau}_{t_{[j]}}|$ represents the maximum change in any of the updated parameters during fine-tuning. Inequality 9 provides an upper bound on the degree of *function localization* of $\boldsymbol{\tau}_t$ obtained via `TaLoS`. Having this quantity equal zero ensures no task interference, as the overall output falls back to $f(\cdot, \boldsymbol{\theta}_0)$ which by definition is *weight disentangled*. Yet, this means that no learning has occurred. Instances of this are when no parameter is updated ($k = 0$) or when only parameters with exactly zero influence ($\eta = 0$) are fine-tuned. Apart from these cases, fine-tuning the least sensitive parameters allows for a minimal increase of this bound while still allowing to learn the task, as even parameters with marginal influence can collectively contribute to task performance (Ben Zaken et al., 2022; Xu et al., 2020; Liao et al., 2023) (in Appendix A.6 we show that `TaLoS` enables learning on par with other PEFT baselines). Indeed, as the model is robust to changes within the flat subspace defined by its least sensitive parameters, learning $\boldsymbol{\tau}_t$ in this subspace ensures minimal impact on the model's output for other tasks as well (we empirically validate this in Figure 3). As detailed in Appendix A.1, $k$ is a hyperparameter controlling the sparsity ratio of $\boldsymbol{c}$, thus, indirectly controlling the degree of function localization. We tuned it at the task level, resulting in optimal sparsity ratios between 90% and 99% (ablation in Appendix A.5).

## 5 EXPERIMENTS

Our experimental evaluation focuses on the established Task Arithmetic framework outlined by Ilharco et al. (2022; 2023), specifically targeting Task Addition and Task Negation, encompassing both language and vision domains. In the following we describe the baselines we compared our `TaLoS` against. Further details regarding the experimental setups, the relevant metrics, the implementation of the experiments, as well as the data and architectures used, are deferred to Appendix A.1. Additionally, in Appendix A.8 we test different model merging schemes on task vectors obtained with `TaLoS`.

**Baselines.** We consider three families of methods as references. (i) **Full fine-tuning** methods aim to produce task vectors $\boldsymbol{\tau}_t$ by fine-tuning all the parameters of the network. Specifically, *Non-linear fine-tuning* (FT) (Ilharco et al., 2022; 2023) minimizes a standard cross-entropy loss, while *Linearized FT* fine-tunes the linearized counterpart of the network, as in Ortiz-Jimenez et al. (2023). (ii) **Post-hoc** methods refine $\boldsymbol{\tau}_t$ after it has been obtained via fine-tuning (as prescribed by the respective methods, we apply these post-hoc approaches on non-linear FT checkpoints). *TIES-Merging* (Yadav et al., 2023) reduces redundancy in $\boldsymbol{\tau}_t$ by magnitude pruning, keeping only the top-$k$ highest magnitude parameters, and addressing sign conflicts when merging task vectors. *TALL Mask / Consensus* (Wang et al., 2024) identifies task-specific parameters in $\boldsymbol{\tau}_t$ by comparing them to the sum of task vectors. It then merges multiple task vectors by using an element-wise OR operation between masks to further identify and remove conflicting parameters. *DARE* (Yu et al., 2024) randomly sparsifies $\boldsymbol{\tau}_t$ to eliminate redundancy and upweights the remaining parameters based on the percentage that was removed. *Breadcrumbs* (Davari & Belilovsky, 2024) reduces redundancy using magnitude pruning and eliminates weight outliers within the retained top-$k$ parameters. Although these methods have been presented for task addition, we also test their ability of handling task negation. (iii) **Parameter-efficient fine-tuning (PEFT)** methods aim to obtain task vectors by efficiently fine-tuning the network, using far fewer resources compared to full fine-tuning. We compare against *L-LoRA* (Tang et al., 2024), which applies linearized low-rank adapters to the $\boldsymbol{Q}$ and $\boldsymbol{V}$ projections in self-attention layers. This approach was specifically designed for Task Arithmetic and offers superior performance over standard LoRA. For sparse fine-tuning, we use *LoTA* (Panda et al., 2024), a method that leverages the Lottery Ticket hypothesis (Frankle & Carbin, 2019) to select the top-$k$ parameters when sparsely fine-tuning the network, making it suitable for model merging.

### 5.1 TASK ARITHMETIC RESULTS

We thoroughly evaluate `TaLoS` on its ability to derive task vectors that enable model editing through simple arithmetic operations on model parameters.

**Task Addition.** In this benchmark, the sum of the task vectors $\sum_t \alpha_t \boldsymbol{\tau}_t$ is added to a pre-trained checkpoint to produce a multi-task model $f(\cdot, \boldsymbol{\theta}_0 + \sum_t \alpha_t \boldsymbol{\tau}_t)$. The success is measured in terms of the maximum average accuracy over the different tasks. As done by Ortiz-Jimenez et al. (2023); Tang

| Method | ViT-B/32 | | ViT-B/16 | | ViT-L/14 | | T5-Small | | T5-Base | | T5-Large | |
|---|---|---|---|---|---|---|---|---|---|---|---|---|
| | Abs. (↑) | Norm. (↑) | Abs. (↑) | Norm. (↑) | Abs. (↑) | Norm. (↑) | Abs. (↑) | Norm. (↑) | Abs. (↑) | Norm. (↑) | Abs. (↑) | Norm. (↑) |
| Pre-trained (Zero-shot) | 47.72 | - | 55.83 | - | 65.47 | - | 55.70 | - | 53.51 | - | 51.71 | - |
| **Full Fine-tuning Methods** | | | | | | | | | | | | |
| Non-linear FT (Ilharco et al., 2023) | 71.25 | 76.94 | 72.85 | 77.17 | 86.09 | 90.14 | **65.04** | 87.98 | 74.20 | 90.63 | 75.37 | 85.25 |
| Linearized FT (Ortiz-Jimenez et al., 2023) | 76.70 | 85.86 | 80.01 | 87.29 | 88.29 | 93.01 | 64.13 | 86.62 | 74.69 | 92.12 | 69.38 | 78.95 |
| **Post-hoc Methods** | | | | | | | | | | | | |
| TIES-Merging (Yadav et al., 2023) | 74.79 | 82.84 | 77.09 | 82.13 | 88.16 | 92.56 | 62.53 | 94.83 | 70.74 | 92.37 | 74.30 | 86.36 |
| TALL Mask / Consensus (Wang et al., 2024) | 74.55 | 80.27 | 74.92 | 79.12 | 86.89 | 90.81 | 63.61 | 95.34 | 73.31 | 91.60 | 77.31 | 87.84 |
| DARE (Yu et al., 2024) | 70.88 | 76.59 | 73.08 | 77.51 | 85.95 | 90.04 | 63.89 | 89.09 | 74.26 | 91.49 | 76.20 | 86.51 |
| Breadcrumbs (Davari & Belilovsky, 2024) | 69.39 | 79.51 | 71.93 | 78.94 | 84.78 | 92.97 | 61.19 | 92.23 | 73.89 | 92.70 | 73.41 | 87.07 |
| **Parameter-efficient Fine-tuning Methods** | | | | | | | | | | | | |
| L-LoRA (Tang et al., 2024) | 78.00 | 86.08 | 80.61 | 85.83 | 87.77 | 91.87 | 60.29 | 94.46 | 68.76 | 91.98 | 72.10 | 87.78 |
| LoTA (Panda et al., 2024) | 64.94 | 74.37 | 79.11 | 83.97 | 87.66 | 91.69 | 64.21 | 87.92 | 74.31 | 92.25 | 75.84 | 88.14 |
| **TaLoS (Ours)** | **79.67** [+1.67] | **90.73** [+4.65] | **82.60** [+1.99] | **91.41** [+4.12] | **88.37** [+0.08] | **95.20** [+2.19] | **65.04** [0.00] | **97.22** [+1.88] | **75.93** [+1.24] | **95.87** [+3.17] | **79.07** [+1.76] | **90.61** [+2.47] |

Table 1: **Task Addition results.** Average absolute accuracies (%) and normalized accuracies (%) of different CLIP ViTs and T5 pre-trained models edited by adding task vectors on each of the downstream tasks. We normalize performance of each method by their single-task accuracy. **Bold** indicates the best results. Underline the second best.

| Method | ViT-B/32 | | ViT-B/16 | | ViT-L/14 | | T5-Small | | T5-Base | | T5-Large | |
|---|---|---|---|---|---|---|---|---|---|---|---|---|
| | Targ. (↓) | Cont. (↑) | Targ. (↓) | Cont. (↑) | Targ. (↓) | Cont. (↑) | Targ. (↓) | Cont. (↑) | Targ. (↓) | Cont. (↑) | Targ. (↓) | Cont. (↑) |
| Pre-trained (Zero-shot) | 47.72 | 63.26 | 55.83 | 68.37 | 65.47 | 75.53 | 55.70 | 45.70 | 53.51 | 45.30 | 51.71 | 45.70 |
| **Full Fine-tuning Methods** | | | | | | | | | | | | |
| Non-linear FT (Ilharco et al., 2023) | 24.04 | 60.36 | 20.36 | 64.79 | 20.61 | 72.72 | 43.06 | 45.47 | 40.06 | 45.16 | 41.54 | 45.49 |
| Linearized FT (Ortiz-Jimenez et al., 2023) | 11.20 | 60.74 | 10.97 | 65.55 | 10.86 | 72.43 | 44.47 | 44.94 | 40.16 | 45.27 | 41.37 | 45.70 |
| **Post-hoc Methods** | | | | | | | | | | | | |
| TIES-Merging (Yadav et al., 2023) | 21.94 | 61.49 | 19.72 | 65.69 | 24.50 | 73.41 | 55.01 | 45.30 | 40.30 | 45.13 | 46.19 | 45.56 |
| TALL Mask / Consensus (Wang et al., 2024) | 23.31 | 60.54 | 20.71 | 65.17 | 22.33 | 73.30 | 43.43 | 45.41 | 40.14 | 45.20 | 41.26 | 45.59 |
| DARE (Yu et al., 2024) | 25.04 | 60.60 | 22.22 | 64.98 | 20.94 | 72.66 | 42.53 | 45.36 | 40.24 | 45.16 | 41.29 | 45.70 |
| Breadcrumbs (Davari & Belilovsky, 2024) | 24.27 | 60.58 | 21.60 | 65.22 | 20.69 | 72.95 | 53.03 | 45.19 | 40.46 | 45.14 | 41.49 | 45.51 |
| **Parameter-efficient Fine-tuning Methods** | | | | | | | | | | | | |
| L-LoRA (Tang et al., 2024) | 17.29 | 60.75 | 19.33 | 65.69 | 19.39 | 73.14 | 55.30 | 45.24 | 51.33 | 45.10 | 48.37 | 45.51 |
| LoTA (Panda et al., 2024) | 21.09 | 61.01 | 17.76 | 65.60 | 22.11 | 73.21 | 54.70 | 45.13 | 40.50 | 45.24 | 44.33 | 45.47 |
| **TaLoS (Ours)** | **11.03** [+0.17] | 60.69 [-0.80] | **10.58** [+0.39] | **66.11** [+0.42] | **10.68** [+0.18] | **73.63** [+0.22] | **39.64** [+2.89] | **45.67** [+0.20] | **38.49** [+1.57] | **45.28** [+0.01] | **37.20** [+4.06] | **45.70** [0.00] |

Table 2: **Task Negation results.** Average minimal accuracy (%) of different CLIP ViTs and T5 pre-trained models edited by subtracting a task vector from a target task while retaining at least 95% of their performance on the control task. We average the minimal accuracy over each of the downstream tasks. **Bold** indicates the best results. Underline the second best.

et al. (2024), we also report the average normalized accuracy over the tasks. The normalization is performed with respect to the single-task accuracies achieved by the model fine-tuned on each task (see Appendix A.1). The results in Table 1 demonstrate the effectiveness of our proposed method across various model scales and modalities. TaLoS consistently outperforms existing approaches, with evident improvements in normalized accuracy of 1.88% to 4.65% over the second best method across all model variants. Such a metric provides insights into the outstanding ability of TaLoS to maximize the benefits of model combination while mitigating interference.

For vision models, TaLoS exhibits strong performance across all scales, with absolute accuracy gains of up to 2.61% over the closest competitor. In NLP, TaLoS maintains its leading position, although the gains are less striking than in vision experiments. Nevertheless, the improvements are particularly pronounced in larger models, suggesting that TaLoS scales well with model size. Notably, TaLoS's performance surpasses both full fine-tuning and post-hoc methods across the board. This suggests that our parameter-efficient approach can achieve superior results while potentially reducing computational costs, a crucial factor when working with large-scale models.

**Task Negation.** In this benchmark a task vector $\boldsymbol{\tau}_t$ is subtracted from the pre-trained checkpoint to reduce the performance on task $t$, producing the model $f(\cdot, \boldsymbol{\theta}_0 - \alpha_t \boldsymbol{\tau}_t)$. By following Ortiz-Jimenez et al. (2023), the success is measured in terms of the maximum drop in accuracy on the forgetting task that retains at least 95% of the accuracy on the control task. Results are averaged over tasks and presented in Table 2. For vision models, TaLoS achieves the lowest target task accuracies while maintaining high control task performance, indicating superior ability to selectively remove targeted task information. For T5 models, all methods, including TaLoS, face significant challenges in Task Negation. The results show a much tighter clustering of performance across different approaches. This suggests that negating specific language tasks without substantially impacting the control task accuracy is inherently more difficult than in vision models. Despite this challenge, TaLoS still manages to achieve the best balance between target and control task performance.

## 5.2 WEIGHT DISENTANGLEMENT AND LOCALIZATION

The improved localization provided by TaLoS seems to play a crucial role in driving effective task arithmetic. Here we delve deeper into this aspect with tailored analyses. First, we assess how well the weight disentanglement property holds. Then, for each training recipe, we evaluate the degree of task component localization on each task.

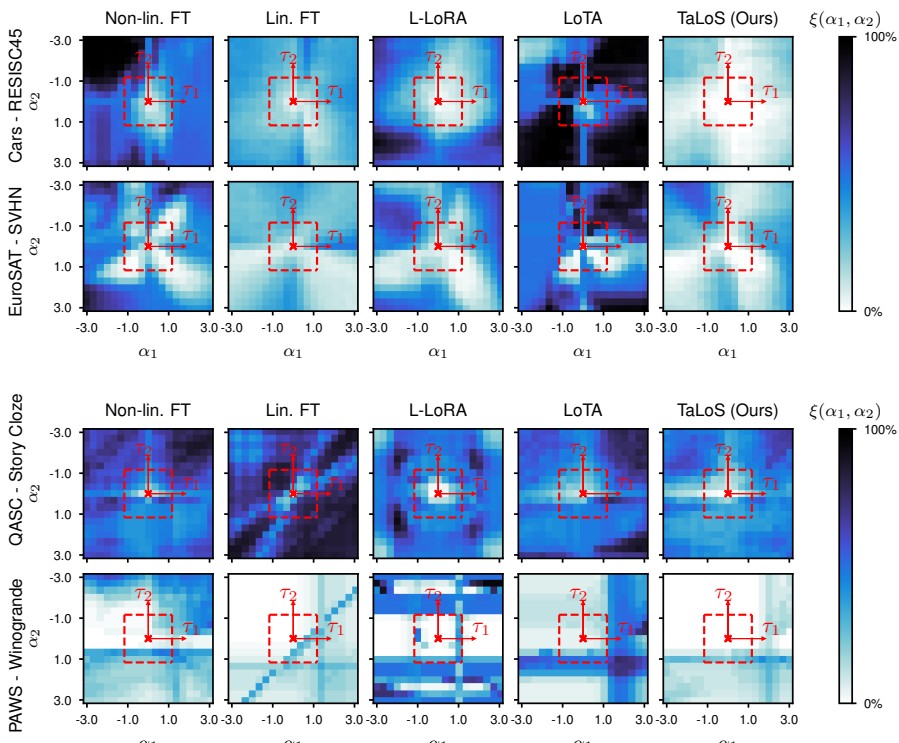

Figure 2: **Visualizing weight disentanglement error.** The heatmaps illustrate the disentanglement error $\xi(\alpha_1, \alpha_2)$ of each fine-tuning strategy on both a CLIP ViT-B/32 model (top) and a T5-Small model (bottom) across two task pairs. Lighter areas highlight regions of the weight space where disentanglement is more pronounced. The red box indicates the search space within which the optimal $\alpha$ values were searched (refer to Appendix A.1). We chose the task pairs to visualize by following Ortiz-Jimenez et al. (2023) for vision and a criterion akin to the one used in Tang et al. (2024) for language.

**Weight disentanglement error visualization.** Ortiz-Jimenez et al. (2023); Tang et al. (2024) proposed to evaluate the *disentanglement error* defined as

$$\xi(\alpha_1, \alpha_2) = \sum_{t=1}^{2} \mathbb{E}_{\boldsymbol{x} \in \mathcal{D}_t} [\text{dist}(f(\boldsymbol{x}, \boldsymbol{\theta}_0 + \alpha_1 \boldsymbol{\tau}_1), f(\boldsymbol{x}, \boldsymbol{\theta}_0 + \alpha_1 \boldsymbol{\tau}_1 + \alpha_2 \boldsymbol{\tau}_2))] \qquad (10)$$

where the *prediction error* $\text{dist}(y_1, y_2) = \mathbb{1}(y_1 \neq y_2)$ is taken as the distance metric. Generally, given a pair $(\alpha_1, \alpha_2)$, the smaller the value of $\xi(\alpha_1, \alpha_2)$ the more weight disentangled a model is. Maintaining a low disentanglement error as $\alpha_1$ and $\alpha_2$ increase provides an even stronger evidence of the weight disentanglement property.

In Figure 2, we report $\xi(\alpha_1, \alpha_2)$ across different fine-tuning strategies for both the CLIP ViT-B/32 and T5-Small models on two task pairs. Overall there is a clear difference in disentanglement patterns between vision and language models. For the latter, the patterns are more consistent across strategies, which may explain why the differences in task arithmetic performance are notable in vision experiments and less pronounced in language experiments (ref. to Tables 1, 2).

By focusing on vision models we observe that Linearized FT, L-LoRA, and our approach demonstrate improved disentanglement (indicated by lighter regions) than non-linear fine-tuning, with our method performing the best overall. We remind that L-LoRA approximate the behavior of Linearized FT via adapters but still lacks to optimize the task localization property. Interestingly, LoTA shows a much lower degree of disentanglement. We remark that this approach selects and updates task-specific parameters while `TaLoS` focuses on task-generic ones and this difference accounts for the observed behavior.

For language, Linearized FT and L-LoRA yield mixed results depending on the pairs of considered tasks. LoTA seems able to improve over non-linearized FT but with different extents across tasks and it is consistently outperformed by `TaLoS`.

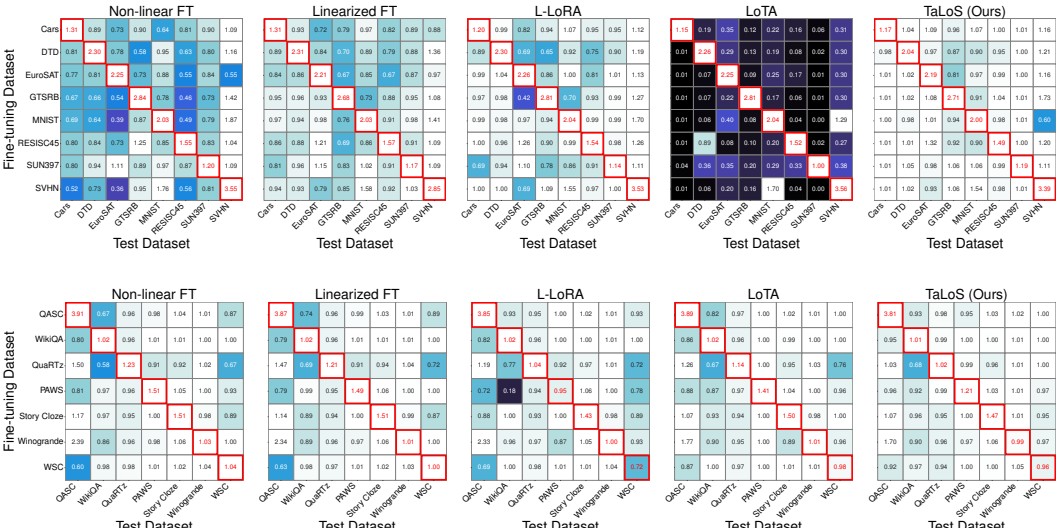

Figure 3: **Function localization.** The heatmaps present the accuracy ratios for fine-tuned models across tasks for CLIP ViT-B/32 (top) and T5-Small (bottom) models. Each row indicates a model fine-tuned on a specific task, with columns representing its performance on different test datasets. Accuracy ratios are normalized by the pre-trained model's performance. Lighter colors indicate better performance, suggesting minimal interference between the fine-tuned model and other tasks' input spaces. The red diagonal highlights each model's test performance on its specific fine-tuning task.

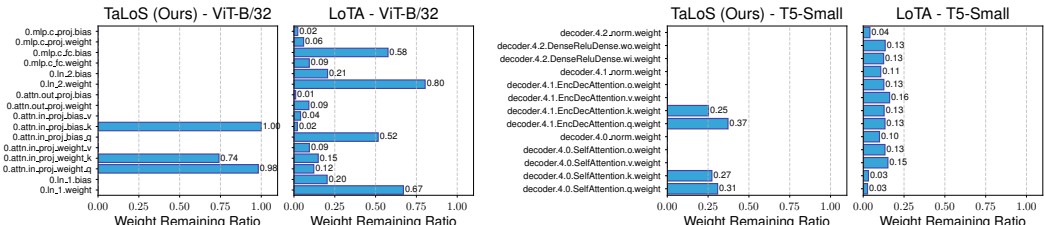

Figure 4: **Visualization of mask calibration.** Percentage of parameters selected for sparse fine-tuning in a transformer block of a ViT-B/32 (left) and a T5-Small (right) models, after our method's mask calibration vs. LoTA's mask calibration, at 90% sparsity. On ViT-B/32, we calibrate the masks on the Cars dataset (Krause et al., 2013), while on T5-Small we use QASC (Khot et al., 2020). Full visualizations of all masked layers are reported in Appendix A.3.

**Function localization.** We experimentally assess the function localization property of `TaLoS` by comparing it with other fine-tuning methods. From the definition in Equation 6, we know that when this property holds, each task activates only for its specific data support. Thus, we should observe an advantage in the prediction output when testing on that task, and the same performance of the pre-trained model for all the others tasks. Figure 3 confirms the expected behavior for `TaLoS` in vision, while the competitors display more interference between tasks, as indicated by darker hues off the diagonal. Interestingly, for NLP tasks all methods exhibit natural function localization, as reflected by the lighter regions in the figure. This provides us the opportunity to remark the importance of extensive model analysis as conclusions drawn from a single domain where linearization is sufficient might be misleading.

## 5.3 WEIGHT SPARSITY STRUCTURE AND EFFICIENCY

**Visualizing task vector masks.** To understand the nature of our sparse fine-tuning approach, we analyze the structure of the masks $c$ calibrated using `TaLoS` and compare it with the ones produced by LoTA. Figure 4 provides a visualization of the layer-wise percentage of parameters selected for sparse fine-tuning in a transformer block of a ViT-B/32 and a T5-Small models. The results reveal distinct patterns in parameter selection between `TaLoS` and LoTA across both models. `TaLoS` exhibits a highly structured selection, predominantly preserving parameters in the multihead self-attention layer, particularly in the $Q$ and $K$ projections. In contrast, LoTA's selection appears more distributed across different layers of the transformer block. Interestingly, our analysis reveals some notable contrasts

| Method | Effective Cost of Fine-tuning | | | | Task Addition | | Task Negation | |
|---|---|---|---|---|---|---|---|---|
| | Forward-Backward Pass Time (s) | Optim. Step Time (s) | Tot. Iteration Time (s) | Peak Memory Usage (GiB) | Abs. (↑) | Norm. (↑) | Targ. (↓) | Cont. (↑) |
| **ViT-B/32** | | | | | | | | |
| Non-linear FT (Ilharco et al., 2023) | $0.3608 \pm 0.0036$ | $\underline{0.0114} \pm 0.0010$ | $0.3722 \pm 0.0037$ | 6.5 | 71.25 | 76.94 | 24.04 | 60.36 |
| Linearized FT (Ortiz-Jimenez et al., 2023) | $0.6858 \pm 0.0042$ | $0.0103 \pm 0.0020$ | $0.6961 \pm 0.0047$ | 10.2 | 76.70 | 85.86 | $\underline{11.20}$ | 60.74 |
| L-LoRA (Tang et al., 2024) | $\underline{0.3270} \pm 0.0076$ | $\mathbf{0.0036} \pm 0.0032$ | $\underline{0.3306} \pm 0.0082$ | $\underline{5.3}$ | $\underline{78.00}$ | $\underline{86.08}$ | 17.29 | $\underline{60.75}$ |
| LoTA (Panda et al., 2024) | $0.3289 \pm 0.0041$ | $0.1269 \pm 0.0050$ | $0.4558 \pm 0.0065$ | 6.8 | 64.94 | 74.37 | 21.09 | **61.01** |
| **TaLoS (Ours)** | $\mathbf{0.1256} \pm 0.0045$ | $0.0388 \pm 0.0040$ | $\mathbf{0.1644} \pm 0.0060$ | **4.7** | **79.67** | **90.73** | **11.03** | 60.69 |
| **ViT-L/14** | | | | | | | | |
| Non-linear FT (Ilharco et al., 2023) | $1.2174 \pm 0.0097$ | $\underline{0.0156} \pm 0.0055$ | $1.2330 \pm 0.0112$ | 18.6 | 86.09 | 90.14 | 20.61 | 72.72 |
| Linearized FT (Ortiz-Jimenez et al., 2023) | $1.6200 \pm 0.0067$ | $0.0262 \pm 0.0082$ | $1.6462 \pm 0.0106$ | 21.3 | $\underline{88.29}$ | $\underline{93.01}$ | $\underline{10.86}$ | 72.43 |
| L-LoRA (Tang et al., 2024) | $0.5153 \pm 0.0077$ | $\mathbf{0.0082} \pm 0.0015$ | $\underline{0.5235} \pm 0.0078$ | $\underline{9.7}$ | 87.77 | 91.87 | 19.39 | 73.14 |
| LoTA (Panda et al., 2024) | $0.8438 \pm 0.0052$ | $0.4449 \pm 0.0074$ | $1.2887 \pm 0.0090$ | 15.4 | 87.66 | 91.69 | 22.11 | $\underline{73.21}$ |
| **TaLoS (Ours)** | $\mathbf{0.1891} \pm 0.0039$ | $0.1372 \pm 0.0036$ | $\mathbf{0.3263} \pm 0.0053$ | **7.8** | **88.37** | **95.20** | **10.68** | **73.63** |
| **T5-Large** | | | | | | | | |
| Non-linear FT (Ilharco et al., 2023) | $0.9047 \pm 0.0068$ | $0.0894 \pm 0.0034$ | $0.9941 \pm 0.0076$ | 30.0 | 75.37 | 85.25 | 41.54 | 45.49 |
| Linearized FT (Ortiz-Jimenez et al., 2023) | $1.7683 \pm 0.0084$ | $0.1170 \pm 0.0060$ | $1.8853 \pm 0.0103$ | 35.1 | 69.38 | 78.95 | $\underline{41.37}$ | **45.70** |
| L-LoRA (Tang et al., 2024) | $\underline{0.7452} \pm 0.0084$ | $\mathbf{0.0136} \pm 0.0029$ | $\underline{0.7588} \pm 0.0089$ | $\underline{18.2}$ | 72.10 | 87.78 | 48.37 | 45.51 |
| LoTA (Panda et al., 2024) | $0.8526 \pm 0.0043$ | $0.3842 \pm 0.0019$ | $1.2368 \pm 0.0047$ | 32.1 | $\underline{75.84}$ | $\underline{88.14}$ | 44.33 | 45.47 |
| **TaLoS (Ours)** | $\mathbf{0.4358} \pm 0.0075$ | $\underline{0.0509} \pm 0.0046$ | $\mathbf{0.4867} \pm 0.0088$ | **12.1** | **79.07** | **90.61** | **37.20** | **45.70** |

Table 3: **Computational cost and memory footprint of fine-tuning.** Average iteration time (in seconds) and peak memory usage (in Gibibytes) of different fine-tuning approaches on CLIP ViT-B/32, ViT-L/14 and T5-Large models, alongside their performance on the task arithmetic benchmark. To improve granularity, we report also the average forward-backward time of a single iteration and the average step time of the optimizer. We separate full fine-tuning methods from parameter-efficient fine-tuning methods. Further details on the resource monitoring process can be found in Appendix A.1. **Bold** indicates the best results. Underline the second best.

with L-LoRA (Tang et al., 2024), a method specifically designed for task arithmetic. While L-LoRA arbitrarily fine-tunes the $Q$ and $V$ projections, our findings suggest that, generally, $Q$ and $K$ play a more significant role in task arithmetic than $V$ in the multihead self-attention layers. Additionally, for CLIP ViT-B/32 biases also seemingly play a crucial role for function localization. This structured sparsity not only provides insights into our method's mask calibration mechanism but also hints at potential efficiency gains, which we explore further in the following.

**Computational cost and memory footprint.** The observed structured sparsity pattern of `TaLoS` suggests that it also provides a highly efficient task arithmetic fine-tuning strategy. To verify it we performed a comparative analysis of the computational cost and memory footprint of `TaLoS` against several fine-tuning methods.

In Table 3 we present the collected time and memory costs with detailed average time (in seconds) for a single training iteration's forward and backward pass. This is separated because approaches like Linearized FT and L-LoRA involve specialized forward passes that require Jacobian-vector products with respect to LoTA and `TaLoS`, which operate similarly to non-linear FT. We also report the time (in seconds) spent by the optimizer updating parameters, as LoTA and `TaLoS` require an additional mask-based element-wise multiplication to prevent updates to certain parameters by masking gradients. Additionally, we provide the total time (sum of these two values) and the peak memory usage (in Gibibytes) recorded during fine-tuning. Overall, the ability to freeze a large number of parameters, thanks to well-structured mask sparsity of our approach improves the total iteration time. Although our method has a slower optimizer step compared to other approaches, the faster forward-backward pass compensates, making `TaLoS` the leading method. In terms of memory usage, the benefits are especially notable for large models, where only a small subset of parameters requires fine-tuning, thus, yielding pronounced savings.

## 6 CONCLUSION

In this work we have proposed `TaLoS`, an efficient and effective strategy to edit pre-trained models in the framework of task arithmetic. We started from the observation that the parameters showing the least variation in the fine-tuning process of a single task are also those minimally relevant for other tasks. Thus, we have leveraged them through a sparse learning process that promotes task localization and avoids task interference. A thorough experimental analysis across vision and language domains confirmed that `TaLoS` yields state-of-the-art results in task addition and negation, showing a significant efficiency advantage over competitors. Moreover, with a tailored set of evaluations we assessed model linearization and function localization properties, providing insights on the inner functioning of our approach.

Overall, we have discussed how preserving the regularities provided by a large scale pre-trained model are sufficient to maintain weight disentanglement and observe beneficial effects in task arithmetic. Future work may investigate whether explicitly enforcing localization constraints during fine-tuning could enhance performance and further advance model editing capabilities.

## REPRODUCIBILITY STATEMENT

We have made significant efforts to ensure the reproducibility of our results. Full implementation details are provided in Appendix A.1. Pseudocode for our algorithm is included in Appendix A.2 to clarify key steps, as well as practical design choices to address potential challenges in implementing our experiments. Additionally, we publicly released our code to further facilitate reproducibility at `https://github.com/iurada/talos-task-arithmetic`.

## ACKNOWLEDGEMENTS

The authors thank the reviewers and area chair for their valuable comments. M.C. also thanks Derek Tam and Colin Raffel for their fruitful discussions and feedback on the early state of this work. L.I. acknowledges the grant received from the European Union Next-GenerationEU (Piano Nazionale di Ripresa E Resilienza (PNRR)) DM 351 on Trustworthy AI. T.T. acknowledges the EU project ELSA - European Lighthouse on Secure and Safe AI. This study was carried out within the FAIR - Future Artificial Intelligence Research and received funding from the European Union Next-GenerationEU (PIANO NAZIONALE DI RIPRESA E RESILIENZA (PNRR) – MISSIONE 4 COMPONENTE 2, INVESTIMENTO 1.3 – D.D. 1555 11/10/2022, PE00000013). This manuscript reflects only the authors' views and opinions, neither the European Union nor the European Commission can be considered responsible for them. We acknowledge the CINECA award under the ISCRA initiative for the availability of high-performance computing resources and support.

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

# A APPENDIX

## A.1 IMPLEMENTATION DETAILS

**Computational resources.** We execute all the vision experiments using ViT-B/32, ViT-B/16, and ViT-L/14 on a machine equipped with two NVIDIA GeForce RTX 2080 Ti (11 GB VRAM), an Intel Core i7-9800X CPU @ 3.80GHz and 64 GB of RAM. For all the language experiments using T5-Small, T5-Base, and T5-Large we employ a machine equipped with a a single NVIDIA A100 SXM (64 GB VRAM), an Intel Xeon Platinum 8358 CPU @ 2.60GHz and 64 GB of RAM.

**Starter code.** We developed our codebase starting from the repositories provided by Ortiz-Jimenez et al. (2023)[3] (based on the code by Ilharco et al. (2022; 2023)[4]) and Yadav et al. (2023)[5], which allow to reproduce the full fine-tuning results (Non-linear FT and Linearized FT). TIES-Merging (Yadav et al., 2023)[5], TALL Mask / Consensus (Wang et al., 2024)[6], DARE (Yu et al., 2024)[7], Breadcrumbs (Davari & Belilovsky, 2024)[8] and LoTA (Panda et al., 2024)[9] provide official implementations of their methods from which we carefully adapted their code to work within the Task Arithmetic framework. L-LoRA (Tang et al., 2024) unfortunately doesn't provide any official implementation, but the guidelines in the paper are sufficient to reproduce their results. To this end, we used the `peft` library (Mangrulkar et al., 2022)[10] for implementing the LoRA modules.

**Hyperparameter selection.** As highlighted by Ortiz-Jimenez et al. (2023), task vectors that perform well in Task Negation tend to exhibit higher degrees of weight disentanglement in Task Addition. This relationship informed our hyperparameter selection strategy. For each method, we cross-validate its hyperparameters on each individual task by leveraging Task Negation performance on a small held-out portion of the training set, as implemented by Ilharco et al. (2023); Ortiz-Jimenez et al. (2023). It's important to note that hyperparameter selection shall not be performed separately for addition and negation, as each choice of hyperparameters yields a unique task vector. Hyperparameter search of each method is carried out according to the guidelines presented in each paper. Specifically, for **post-hoc** methods, the sparsity ratio is searched in the set $k \in \{0.1, 0.2, ..., 0.9, 0.95, 0.99\}$. Furthermore, for TALL Mask / Consensus (Wang et al., 2024) we also tune the *consensus threshold* in the set $\{0, ..., T\}$, where $T$ is the number of tasks. For Breadcrumbs (Davari & Belilovsky, 2024) we also tune the percentage of top-$k$ parameters considered outliers, using values from the set $\{0.8, 0.9, 0.95, 0.99, 0.992, 0.994, ..., 0.999\}$. Regarding **parameter-efficient fine-tuning** methods, when using L-LoRA (Tang et al., 2024) we progressively reduce its rank $r \in \{512, 256, 128, 64, 32, 16, 8\}$. While, for LoTA (Panda et al., 2024) and our method we tune sparsity at the task level using values in the set $\{0.1, 0.2, ..., 0.9, 0.95, 0.99\}$. Regarding the amount of data used to perform mask calibration on each task, we align with Panda et al. (2024) by using the validation split as it accounts for the 10% of the total training data. For LoTA, we set the number of iterations for mask calibration so to match the number of mask calibration rounds used by our method (further details at Section A.2). This ensures that the drop in performance is negligible with respect to using the full training split while significantly reducing the computational overhead.

**Datasets & Tasks.** In line with what introduced in Ilharco et al. (2022; 2023); Ortiz-Jimenez et al. (2023), our vision experiments consider image classification across various domains. We adhere to the proposed experimental setup by utilizing eight datasets: Cars (Krause et al., 2013), DTD (Cimpoi et al., 2014), EuroSAT (Helber et al., 2019), GTSRB (Stallkamp et al., 2011), MNIST (LeCun, 1998), RESISC45 (Cheng et al., 2017), SUN397 (Xiao et al., 2016) and SVHN (Netzer et al., 2011).

For the natural language processing (NLP) experiments, we follow the methodology outlined in Yadav et al. (2023), incorporating seven prescribed datasets: three regarding question answering (QASC (Khot et al., 2020), WikiQA (Yang et al., 2015) and QuaRTz (Tafjord et al., 2019)), one for paraphrase identification (PAWS (Zhang et al., 2019)), one focusing on sentence completion (Story Cloze (Sharma et al., 2018)) and two for coreference resolution (Winogrande (Sakaguchi et al., 2021)

---

[3]https://github.com/gortizji/tangent_task_arithmetic
[4]https://github.com/mlfoundations/task_vectors
[5]https://github.com/prateeky2806/ties-merging
[6]https://github.com/nik-dim/tall_masks
[7]https://github.com/yule-BUAA/MergeLM
[8]https://github.com/rezazzr/breadcrumbs
[9]https://github.com/kiddyboots216/lottery-ticket-adaptation
[10]https://github.com/huggingface/peft

and WSC (Levesque et al., 2012)). Concerning Task Negation, we align with Ortiz-Jimenez et al. (2023) and consider ImageNet (Deng et al., 2009) as the control dataset for vision experiments, while for NLP, we utilize RTE (Dagan et al., 2005), as it provides a distinct task (*i.e.* natural language inference) with respect to the others considered for the NLP experiments.

**Architectures & Pre-trained models.** By following Ilharco et al. (2023); Ortiz-Jimenez et al. (2023); Yadav et al. (2023), on vision experiments, we use three variants of CLIP (Radford et al., 2021) with ViT-B/32, ViT-B/16, and ViT-L/14 models (Dosovitskiy et al., 2021). Regarding the NLP experiments, we employ T5-Small, T5-Base, and T5-Large models (Raffel et al., 2020).

**Fine-tuning details.** All fine-tuning experiments on vision adhere to the training protocol outlined by Ilharco et al. (2022; 2023); Ortiz-Jimenez et al. (2023), with minor modifications made to the training code to accommodate the additional baselines and our method. Specifically, we fine-tune all datasets starting from the same CLIP pre-trained checkpoint, which is obtained from the `open_clip` repository (Gadre et al., 2024). Each model is fine-tuned for 2,000 iterations with a batch size of 128, a learning rate of $10^{-5}$, and a cosine annealing learning rate schedule with 200 warm-up steps. We use the AdamW optimizer (Loshchilov & Hutter, 2019). Following Ilharco et al. (2022), the weights of the classification layer, which are derived from encoding a standard set of zero-shot template prompts for each dataset, are frozen during fine-tuning. Freezing this layer ensures no additional learnable parameters are introduced and does not negatively affect accuracy (Ilharco et al., 2022). Regarding the language experiments, we aligned with Yadav et al. (2023); Ilharco et al. (2023) and utilized three variants of the T5 model (Raffel et al., 2020), namely T5-Small, T5-Base, and T5-Large, with training conducted for a maximum of 75,000 steps. We employed an effective training batch size of 1024, with a learning rate of $10^{-4}$. To prevent overfitting, we implemented an early stopping mechanism with a patience threshold of 5. During training, we used `bfloat16` and the maximum sequence length was set to 128. Evaluation is carried out by performing rank classification, where the model's log probabilities for all possible label strings are ranked. The prediction is considered correct if the highest-ranked label corresponds to the correct answer.

**Disentanglement error heatmaps.** As prescribed by Ortiz-Jimenez et al. (2023), we produce the weight disentanglement visualizations of Figure 2 by computing the value of the disentanglement error $\xi(\alpha_1, \alpha_2)$ on a $20 \times 20$ grid of equispaced values in $[-3, 3] \times [-3, 3]$. Estimations are carried out on a random subset of 2,048 test points for each dataset.

**Tuning of $\alpha$ in Task Arithmetic experiments.** As outlined in Ilharco et al. (2023); Ortiz-Jimenez et al. (2023), we employ a single coefficient, denoted as $\alpha$, to adjust the size of the task vectors used to modify the pre-trained models (*i.e.* $\alpha_1 = \alpha_2 = ...\alpha_t$). For both the task addition and task negation benchmarks, following fine-tuning, we evaluate different scaling coefficients from the set $\alpha \in \{0.0, 0.05, 0.1, ..., 1.0\}$ and select the value that achieves the highest target metric on a small held-out portion of the training set, as specified in Ilharco et al. (2023); Ortiz-Jimenez et al. (2023). To account for the lower norm of task vectors obtained via sparse fine-tuning (LoTA and `TaLoS`) we extend this range by $\times 1/(1 - k)$ where $k$ is the sparsity ratio of the task vector. Specifically, we aim to maximize the *normalized average accuracy* for Task Addition and ensure the minimum *target accuracy* for Task Negation while maintaining at least 95% of the original accuracy of the pre-trained model on the control task. The tuning of $\alpha$ is performed independently for each method.

**Measuring computational costs and memory footprint.** The timings in Table 3 are obtained using the `perf_counter` clock from Python's `time` module. We monitored memory footprint using the NVIDIA `nvml` library [11]. All measurements are obtained during fine-tuning, with the very same setup explained in the fine-tuning details. Then, for each method, the mean and standard deviation of the timings are computed over all iterations of all tasks. Peak memory usage, instead, is taken as the maximum over all tasks. Memory usage is recorded at regular intervals of 1 second, starting from the first forward pass and ending when the training loop breaks.

**Normalized accuracy calculation in Task Addition.** *Normalized accuracy* is computed by taking the average of the normalized individual accuracies over the $T$ tasks. Given a task $t$, the normalized individual accuracy for $t$ is computed by taking the accuracy of the multi-task fused model on $t$ and dividing it by the single-task accuracy that the fine-tuned checkpoint obtained on $t$ before being fused. Formally,

$$\text{Normalized Accuracy} = \frac{1}{T} \sum_{t=1}^{T} \frac{\text{Accuracy}[f(\mathcal{D}_t, \boldsymbol{\theta}_0 + \sum_{t'}^{T} \alpha_{t'} \boldsymbol{\tau}_{t'})]}{\text{Accuracy}[f(\mathcal{D}_t, \boldsymbol{\theta}_0 + \alpha_t \boldsymbol{\tau}_t)]} \tag{11}$$

---

[11]https://docs.nvidia.com/deploy/nvml-api/

---

**Algorithm 1:** `TaLoS` to obtain task vectors

---

**Input** : Pre-trained model $\boldsymbol{\theta}_0 \in \mathbb{R}^m$, neural network $f(\boldsymbol{x}, \boldsymbol{\theta}) \triangleq \log p_{\boldsymbol{\theta}}(y|\boldsymbol{x})$, task dataset $\mathcal{D}_t$, final sparsity $k$, number of rounds $R$, number of epochs $E$, learning rate $\gamma$, loss function $\mathcal{L}$
**Output** : Task vector $\boldsymbol{\tau}_t \in \mathbb{R}^m$ for performing task arithmetic

1   *// Calibrate sparse fine-tuning mask*
2   $\boldsymbol{c} \leftarrow \mathbb{1}$          ▷ Initialize weight mask to all 1s
3   **for** $r = 1, 2, ..., R$ **do**
4     $p \leftarrow k^{(r/R)}$          ▷ Compute the current sparsity at round $r$
5     $\boldsymbol{s} \leftarrow \boldsymbol{0}$          ▷ Initialize parameter-wise scores to all 0s
6     *// Compute diagonal FIM score according to Equation 7*
7     **for** $\boldsymbol{x} \in \mathcal{D}_t$ **do**
8       $y \sim p_{(\boldsymbol{c} \odot \boldsymbol{\theta}_0)}(y|\boldsymbol{x})$          ▷ Sample from output distribution of the model
9       $\boldsymbol{s} \leftarrow \boldsymbol{s} + [\nabla_{\boldsymbol{\theta}} \log p_{(\boldsymbol{c} \odot \boldsymbol{\theta}_0)}(y|\boldsymbol{x})]^2$          ▷ Update scores on current example & sampled $y$
10     **end**
11     *// Update $\boldsymbol{c}$ to retain only the bottom-k parameters*
12     $\hat{\boldsymbol{s}} \leftarrow$ `sort_descending(`$\boldsymbol{s}$`)`          ▷ Sorted scores in descending order
13     $p \leftarrow \lfloor p \cdot m \rfloor$          ▷ compute bottom-$p$ threshold index
14     **for** $j = 1, 2, ..., m$ **do**
15       **if** $\boldsymbol{s}_{[j]} - \hat{\boldsymbol{s}}_{[p]} > 0$ **then**
16         $\boldsymbol{c}_{[j]} \leftarrow 0$          ▷ Set the mask of the $j$-th parameter to zero
17       **end**
18     **end**
19   **end**
20   *// Sparse fine-tuning, starting from $\boldsymbol{\theta}_0$ and obtaining $\boldsymbol{\theta}_t^{\star}$*
21   **for** $epoch = 1, 2, ..., E$ **do**
22     **for** $(\boldsymbol{x}, y) \in \mathcal{D}_t$ **do**
23       $\boldsymbol{\theta} \leftarrow \boldsymbol{\theta} - \gamma [\boldsymbol{c} \odot \nabla_{\boldsymbol{\theta}} \mathcal{L}(f(\boldsymbol{x}, \boldsymbol{\theta}), y)]$          ▷ Update rule, mask gradients with $\boldsymbol{c}$
24     **end**
25   **end**
26   $\boldsymbol{\tau}_t \leftarrow \boldsymbol{\theta}_t^{\star} - \boldsymbol{\theta}_0$          ▷ Compute final task vector for task $t$
27   **return** $\boldsymbol{\tau}_t$

---

## A.2   DETAILS ON MASK CALIBRATION & COMPUTATIONAL OVERHEAD

Sparse fine-tuning prescribes to mask gradients when updating the model parameters. Thus, it is foundational that the mask is correctly calibrated before training. We mask only Linear, Attention, LayerNorm, and Convolutional layers (Kwon et al., 2022). Embedding layers and final projection layers are kept frozen. Furthermore, following standard procedures in Pruning-at-Initialization (PaI) (Tanaka et al., 2020; Wang et al., 2023), we iteratively refine the mask in multiple rounds to obtain better estimates from the mask calibration procedures. In detail, at each round, we select the bottom-$p$ parameters (according to our score, detailed in Section 4) and we exponentially increase the current sparsity $p$. We repeat this process until we reach the target sparsity $k$. For the sake of major clarity, we report in Algorithm 1 the pseudocode for our procedure, encompassing both mask calibration and sparse fine-tuning. We remark that the choice of the bottom-$k$ values may lead to *layer collapse* (Tanaka et al., 2020), namely, removing all parameters in a layer, disrupting the information flow in the network. To face this problem, we set $\boldsymbol{c}$ to some positive value close to zero (*e.g.* 0.01) and we don't include in the ranking those entries that are already soft-masked. This ensures that we are not changing the nature of our estimation, while countering the possibility of disrupting gradient flow in the network, during calibration.

Unfortunately, mask calibration introduces some amount of overhead before training. It is of paramount importance that such overhead doesn't hinder the computational gains obtained during fine-tuning.

**Time overhead.** The time spent for a single iteration of mask calibration is comparable to that of a single forward-backward iteration of non-linear fine-tuning (refer to Table 3). Our mask calibration process typically employs an average of 10 iterations per round, with satisfactory results already observed at just 4 rounds (*i.e.*, approximately 40 iterations total, we use the same batch size for mask calibration as for fine-tuning). Given that fine-tuning generally requires around 2,000 iterations for vision experiments and substantially more for language tasks, we argue that the time overhead introduced by our mask calibration is negligible.

| Method | Avgerage Execution Time (s) | | | Peak Memory Usage (GiB) | | | Task Addition | | Task Negation | |
|---|---|---|---|---|---|---|---|---|---|---|
| | Mask | Train | Total | Mask | Train | Overall | Abs. (↑) | Norm. (↑) | Targ. (↓) | Cont. (↑) |
| Non-linear FT (Ilharco et al., 2023) | - | 2479.99 | 2479.99 | - | 18.6 | 18.6 | 86.09 | 90.14 | 20.61 | 72.72 |
| Linearized FT (Ortiz-Jimenez et al., 2023) | - | 3311.77 | 3311.77 | - | 21.3 | 21.3 | 88.29 | 93.01 | 10.86 | 72.43 |
| L-LoRA (Tang et al., 2024) | - | 1053.07 | 1053.07 | - | 9.7 | 9.7 | 87.77 | 91.87 | 19.39 | 73.14 |
| LoTA (Panda et al., 2024) | 51.84 | 2592.40 | 2644.24 | 12.9 | 15.4 | 15.4 | 87.60 | 91.89 | 22.02 | 73.22 |
| **TaLoS (Ours)** | 63.04 | **656.23** | **719.27** | 7.8 | 7.8 | 7.8 | **88.40** | **95.19** | 10.63 | **73.55** |

Table 4: **Computational cost and memory footprint of mask calibration and fine-tuning.** Average time (in seconds) and peak memory usage (in Gibibytes) of mask calibration and fine-tuning approaches on CLIP ViT-L/14, alongside their performance on the task arithmetic benchmark. For both LoTA and TaLoS, we used batch size 128 for 40 iterations (in detail, 10 iterations per round for TaLoS, with 4 rounds total). We employ gradient checkpointing during mask calibration. Further details on the resource monitoring process can be found in Appendix A.1. **Bold** indicates the best results. Underline the second best.

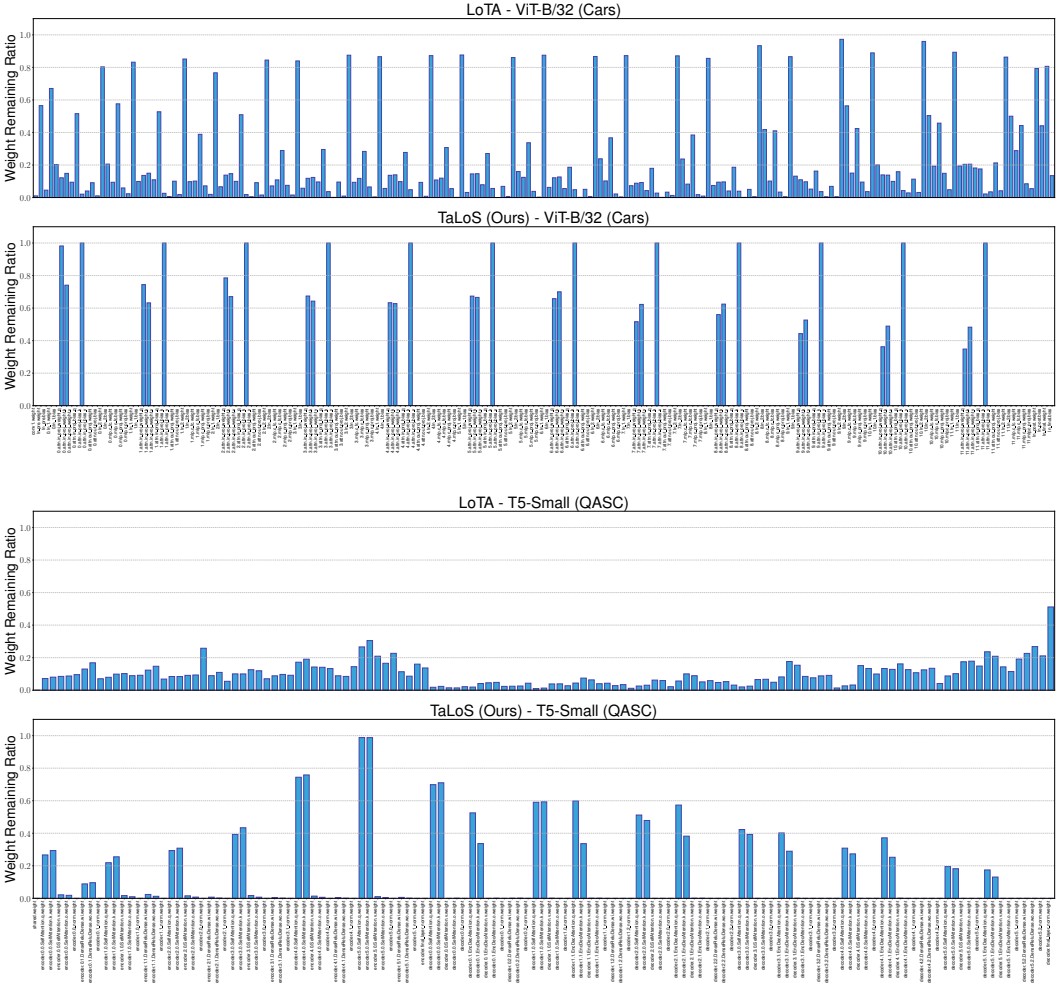

Figure 5: **Visualization of mask calibration.** Percentage of parameters selected for sparse fine-tuning in a ViT-B/32 (top) and a T5-Small (bottom) models, after our method's mask calibration vs. LoTA's mask calibration, at 90% sparsity. On ViT-B/32, we calibrate the masks on the Cars dataset (Krause et al., 2013), while on T5-Small we use QASC (Khot et al., 2020).

**Memory overhead.** The memory cost of each mask calibration iteration is equivalent to that of each training iteration in non-linear fine-tuning. While we have not implemented any specific mechanism to reduce the memory footprint for calculating gradients (used as scores) during mask calibration, there are several approaches available to achieve this. Most of these methods involve estimating gradients using zeroth-order information (Hinton, 2022; Malladi et al., 2023a; Sung et al., 2024), which allows

to trade off speed for reduced memory usage by approximating gradients through multiple forward passes, eliminating the need to store computational graphs for automatic differentiation. Alternatively, gradient checkpointing (Chen et al., 2016) is another practical solution.

To further clarify the overall computational cost of `TaLoS`, encompassing both mask calibration and sparse fine-tuning, we provide a comparison in Table 4 of the timings in seconds (averaged over the 8 vision tasks) and the peak memory usage in Gibibytes of mask calibration and fine-tuning on a CLIP ViT-L/14. The results show that mask calibration time is approximately the same for `TaLoS` and LoTA, however, the costs in terms of memory are very different (LoTA requires storing optimizer states). Regarding total time, we recover what was presented in Table 3, highlighting the beneficial effect of the highly structured sparsity of `TaLoS` on fine-tuning. The task arithmetic results are in line with Tables 1, 2, with no detrimental effect given by the usage of gradient checkpointing.

## A.3 FULL MASK CALIBRATION VISUALIZATIONS

For the sake of completeness, we provide a full visualization in Figure 5 of the masks obtained after calibration with `TaLoS` and LoTA. As shown, a repeating sparsity pattern emerges for our method across each transformer block. Notably, `TaLoS` consistently identifies only the **Q** and **K** parameters for fine-tuning, demonstrating a more structured behavior. In contrast, the mask generated by LoTA appears far more unstructured, with no clear pattern across the blocks.

## A.4 ANALYZING THE FINE-TUNING BEHAVIOR

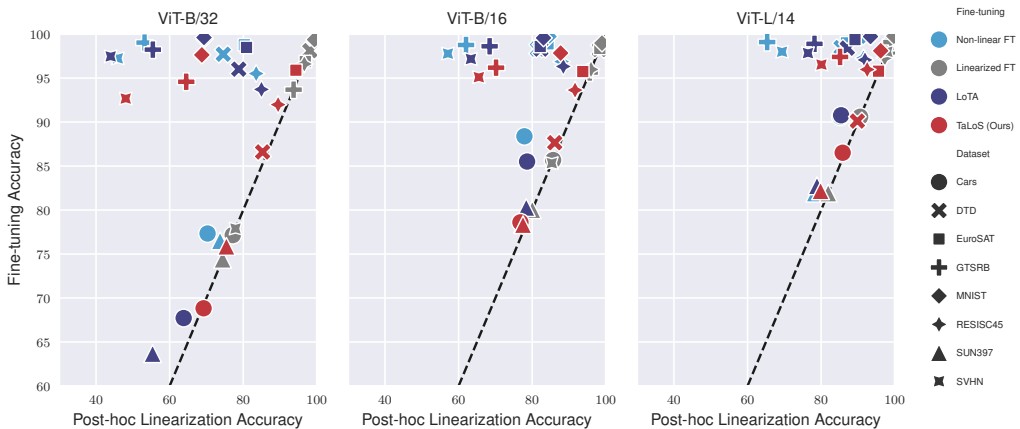

Figure 6: **Testing linearized behavior.** Single-task accuracies of different fine-tuning strategies, each used to obtain their corresponding task vectors $\tau_t$, and the accuracy of their post-hoc linearization $f_{\text{lin}}(\cdot, \theta_0 + \tau_t)$. Different colors represent distinct fine-tuning strategies, while different markers indicate different tasks. Points that lie on the bisector (black dashed line) indicate that the fine-tuning process exhibited linearized behavior.

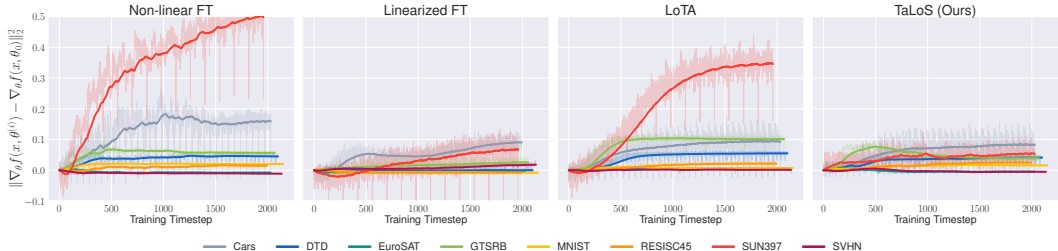

Figure 7: **Change in parameter sensitivity throughout fine-tuning.** We visualize the average relative change in the output derivative of the parameters of a CLIP ViT-B/32 model when fine-tuned using different approaches. The starting point is the same for all methods.

We provide an empirical validation on the linear fine-tuning regime of our `TaLoS` (*i.e.* the change in the network output can be well-approximated by its first-order Taylor expansion around $\theta_0$). As discussed by Ortiz-Jimenez et al. (2023), a cheap test consists of performing post-hoc linearization of

the fine-tuned model around $\boldsymbol{\theta}_0$ and checking whether the performance produced by such a linearized model matches that of the original fine-tuned model. We use this approach and report the results in Figure 6. The scatter plots compare the fine-tuning accuracy against the post-hoc linearization accuracy for various tasks and fine-tuning strategies across different ViT architectures. Our method, TaLoS, consistently demonstrates linearized behavior during fine-tuning for most tasks, as evidenced by its proximity to the bisector line. This supports our claim that sparse fine-tuning, which both TaLoS and LoTA employ, inherently promotes the emergence of linearized behavior during fine-tuning. Interestingly, while TaLoS exhibits this property across a wide range of tasks, LoTA does not consistently demonstrate the same level of linearized behavior. This discrepancy can be attributed to differences in parameter selection, as discussed in the next paragraph, closely matching what happens during linearized fine-tuning. It's worth noting that linearized behavior may arise for various fine-tuning strategies, but its occurrence depends on the interaction between the task and pre-training (Malladi et al., 2023b). For instance, tasks such as GTSRB (Stallkamp et al., 2011), MNIST (LeCun, 1998), and SVHN (Netzer et al., 2011) do not exhibit fine-tuning in the linear regime, hinting at a potential mismatch with the pre-training, as evidence suggests (Radford et al., 2021).

To further test the fine-tuning regime, we examine the evolution of parameter sensitivity during fine-tuning across different methods, as depicted in Figure 7. Inspired by Malladi et al. (2023b), we measure the average change in sensitivity as $\mathbb{E}_{\boldsymbol{x}}[\|\nabla_{\boldsymbol{\theta}} f(\boldsymbol{x}, \boldsymbol{\theta}^{(i)}) - \nabla_{\boldsymbol{\theta}} f(\boldsymbol{x}, \boldsymbol{\theta}_0)\|_2^2]$ at each $i$-th training step, with $\boldsymbol{x}$ from a small subset of 2,048 examples from $\mathcal{D}_t$. Notably, for TaLoS, the gradient $\nabla_{\boldsymbol{\theta}} f(\boldsymbol{x}, \boldsymbol{\theta})$ remains almost unchanged throughout training, closely mirroring the behavior of linearized fine-tuning. In contrast, LoTA diverges from this pattern, behaving more in line with non-linear fine-tuning. This phenomenon reinforces our claim that our method fine-tunes in the linearized regime, as maintaining a constant $\nabla_{\boldsymbol{\theta}} f(\boldsymbol{x}, \boldsymbol{\theta})$ during fine-tuning is critical for operating in the linearized regime (Malladi et al., 2023b).

## A.5 ABLATIONS ON MASK SPARSITY RATIO

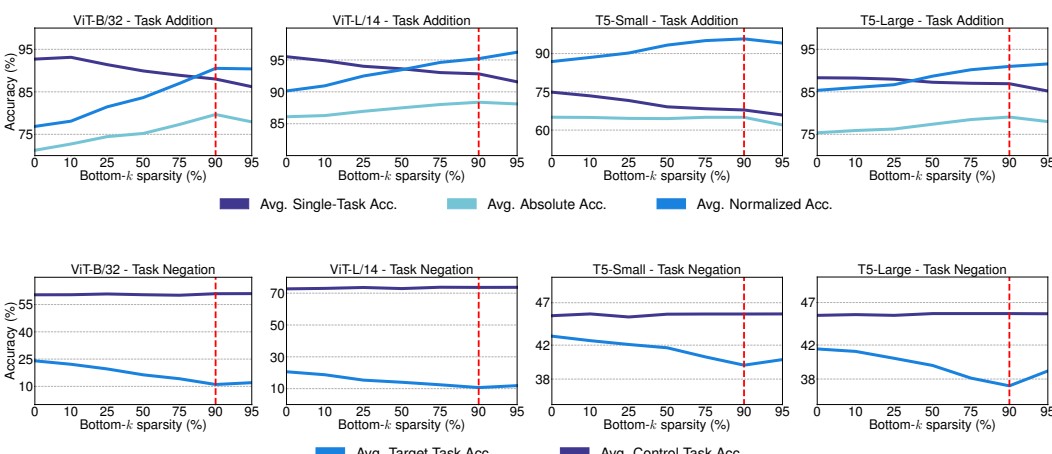

Figure 8: **Effect of the choice of $k$ in TaLoS.** Results of hyperparameter tuning of $k$ in TaLoS for task addition and negation on both vision and language. Note that we tune $k$ indirectly by controlling its value via the sparsity ratio. For **task addition** (top) we report the average single-task accuracy (before addition), absolute and normalized accuracies (after addition). For **task negation** (bottom) we report average target and control accuracies (after negation).

For a clear understanding of the effect of sparsity on TaLoS, we report in Figure 8 the task arithmetic performance achieved by TaLoS, while varying the sparsity level. At 0% sparsity, we recover full (non-linear) fine-tuning results. Increasing the sparsity improves the task arithmetic performance, while slightly decreasing the average single-task accuracy, as fewer parameters are updated during fine-tuning. Optimal values for absolute accuracy (in task addition) and target accuracy (in task negation) are observed for a sparsity level of 90% across a variety of models. After 90% sparsity there is a slight drop in both task arithmetic and single-task performance, making such sparsity levels not ideal. Intuitively, if the fine-tuning involves too little weight the resulting entries in the task vector will be mostly zero, reducing the ability to perform task arithmetic effectively. We can conclude that, like other parameter-efficient fine-tuning methods, our approach trades some single-task performance

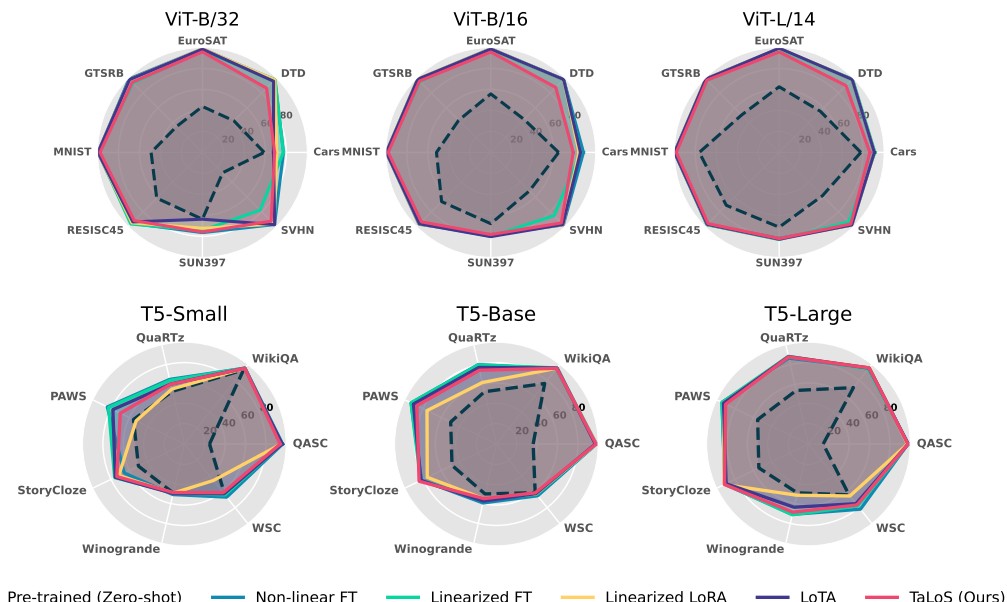

Figure 9: **Task performance after fine-tuning.** Single-task accuracies obtained by different fine-tuning approaches across vision and language experiments. Results are displayed for three model sizes of CLIP ViT (B/32, B/16, L/14) and T5 (Small, Base, Large), with outer edges representing higher accuracy. The dashed line represents the accuracies before fine-tuning.

for parameter efficiency. But this trade-off allows also for superior task arithmetic capabilities for `TaLoS` (Tables 1, 2) while maintaining competitive single-task accuracy, especially for larger models where the performance drop becomes negligible (Figure 9).

### A.6 SINGLE-TASK PERFORMANCE OF FINE-TUNING METHODS

In this analysis we focus on discussing the single-task performance of `TaLoS` before task addition. To this goal, we compare in Figure 9 the accuracies obtained by `TaLoS` (at 90% sparsity) vs. the other fine-tuning strategies. In almost all cases `TaLoS` achieves approximately the same performance of full fine-tuning methods (Non-linear FT and Linearized FT), occasionally improving over Linearized FT (ViT-B/32 on SVHN), which is remarkable, as `TaLoS` updates only a very small subset of parameters, while full fine-tuning (both linearized and non-linear) updates the whole set of model parameters. Furthermore, compared with parameter-efficient fine-tuning methods, which allows for a truly fair comparison (the parameter count is the same across methods), almost always `TaLoS` improves with respect to Linearized LoRA and matches the performance of LoTA. However, we remark that the task arithmetic performance of `TaLoS` is much higher than the latter (see Tables 1, 2).

### A.7 ADDITIONAL EVIDENCE ON THE PARAMETER-SHARING PHENOMENON

In this section, we provide additional validation of the phenomenon observed in our motivating example, namely that insensitive parameters are consistently shared across tasks. First, we revisit the relationship between parameter sensitivity and the Fisher Information matrix (FIM) Fisher (1922), highlighting why the FIM serves as a suitable tool for conducting sensitivity analysis. Next, we present further experimental evidence to support the findings of Section 4.1. Specifically, instead of pruning the least sensitive parameters, we analyze the effect of perturbing them and subsequently examine whether masks calibrated on different tasks exhibit significant similarity.

**Parameter sensitivity analysis and connection to Fisher Information.** Applying a perturbation $\theta_0' \leftarrow \theta_0 + \delta\theta_0$ to a subset of the pre-trained weights $\theta_0$ and observing no change in the output $f(x, \theta_0') \approx f(x, \theta_0)$ intuitively means that those weights have low sensitivity to the task. So, pruning or randomizing them would not affect input-output behavior. However, there may be a problem in assessing sensitivity via extreme randomizations/perturbations: if "extreme" randomization refers to very high magnitude perturbations (perhaps, additive), then such perturbations will not be suitable to assess the sensitivity of the parameters, as this could potentially move the current solution

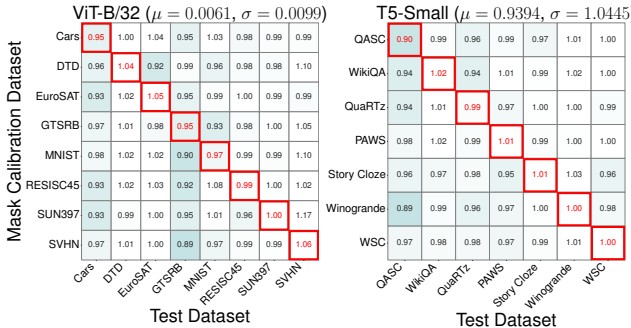

Figure 10: **Perturbing parameters with low sensitivity.** The heatmaps illustrate the effect of perturbing the parameters with the lowest sensitivity (measured by $[F_{[j,j]}(\boldsymbol{\theta}_0, \mathcal{D}_t)]_{j=1}^m$) on different tasks across various pre-trained models. Each grid compares the accuracy ratios for models after pruning, with the rows representing the task $\mathcal{D}_t$ used to identify the parameters with the lowest sensitivity and the columns showing the model's performance on each task after pruning those parameters. The accuracy ratios are normalized by the model's performance before perturbation. The average magnitude $\mu$ and standard deviation $\sigma$ across perturbed parameters, prior to applying noise are also reported. The ratio of perturbed parameters (10%) is chosen based on the experiment of Figure 1.

(parametrized by $\boldsymbol{\theta}_0 \in \mathbb{R}^m$) away from the current local optimum, to a distinct region of the loss landscape. Indeed, sensitivity analysis generally refers to "robustness to small perturbation". This concept, alongside how to perform proper sensitivity analysis on the parameters of a neural network, has been formalized by a rich literature dedicated to applications of information geometry (Amari, 1996; Chaudhry et al., 2018; Pascanu & Bengio, 2013). Specifically, as shown by Chaudhry et al. (2018); Pascanu & Bengio (2013), to assess the influence of each weight on the output of a network, we can use the Kullback-Leibler (KL) divergence between the output distribution induced by the original network ($p_{\boldsymbol{\theta}_0}$) vs. the one induced by the perturbed network ($p_{\boldsymbol{\theta}_0 + \delta\boldsymbol{\theta}}$). Mathematically, assuming $\delta\boldsymbol{\theta} \to 0$ (a small perturbation),

$$D_{\mathrm{KL}}(p_{\boldsymbol{\theta}_0} \| p_{\boldsymbol{\theta}_0 + \delta\boldsymbol{\theta}}) = \frac{1}{2}\delta\boldsymbol{\theta}^\top F(\boldsymbol{\theta}_0)\delta\boldsymbol{\theta} + \mathcal{O}(\|\delta\boldsymbol{\theta}\|^3) \ .$$

The KL divergence is zero if the perturbation doesn't affect the output, revealing that the modified weights are not influential for the output. It is larger than zero otherwise. Here $F(\boldsymbol{\theta}_0) \in \mathbb{R}^{m \times m}$ is the Fisher Information matrix (FIM) (Fisher, 1922; Amari, 1996). It is a positive semi-definite symmetric matrix defined as,

$$F(\boldsymbol{\theta}_0) = \mathbb{E}_{\boldsymbol{x}}[\mathbb{E}_{y \sim p_{\boldsymbol{\theta}_0}(y|\boldsymbol{x})}[\nabla_{\boldsymbol{\theta}} \log p_{\boldsymbol{\theta}_0}(y|\boldsymbol{x})\nabla_{\boldsymbol{\theta}} \log p_{\boldsymbol{\theta}_0}(y|\boldsymbol{x})^\top]] \ .$$

It can be used to relate the changes in the parameters to the changes in the outputs, effectively implementing a proper sensitivity analysis of the parameters of a neural network by studying the magnitude of its diagonal elements, as they represent the sensitivity of each parameter (Chaudhry et al., 2018; Pascanu & Bengio, 2013; Matena & Raffel, 2022). Formally, for each parameter $j \in 1, ..., m$, its corresponding entry on the diagonal of the FIM has value

$$F_{[j,j]}(\boldsymbol{\theta}_0) = \mathbb{E}_{\boldsymbol{x}}[\mathbb{E}_{y \sim p_{\boldsymbol{\theta}_0}(y|\boldsymbol{x})}[\nabla_{\boldsymbol{\theta}_{[j]}} \log p_{\boldsymbol{\theta}_0}(y|\boldsymbol{x})]^2] \ .$$

The higher this value, the more the model will be affected by the $j$-th parameter changes.

**Perturbing the least sensitive parameters.** We repeat in Figure 10 the experiment of Figure 1, but by adding noise distributed as $\mathcal{N}(0, 2\sigma I)$ to the bottom-10% of parameters, instead of pruning them. $\sigma$ is the standard deviation of the parameters, previous to perturbation. The results align with the analysis reported in Figure 1, highlighting the stability of these parameters across tasks.

**Measuring masks intersections across tasks.** Additionally, in Figure 11 we provide further evidence about the overlap of low-sensitivity parameters across tasks. For each parameter, we compute the mean Intersection over Union (mIoU) of masks, between each task pair: starting from pre-trained parameters $\theta_0$, we predict the mask on task $t$ and then check its intersection over union against the mask predicted on task $t'$ (which acts as a ground truth). A mIoU of 1 signals perfect mask overlap

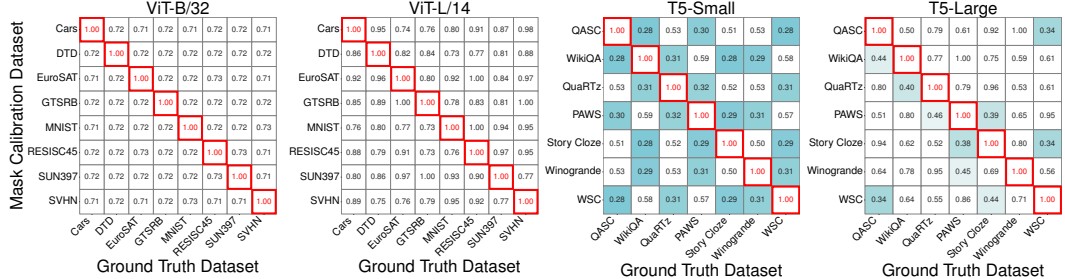

Figure 11: **Masks intersections of low sensitivity parameters.** The heatmaps illustrate the mean Intersection over Union (mIoU) between masks pairs of the lowest sensitivity parameters (measured by $[F_{[j,j]}(\boldsymbol{\theta}_0, \mathcal{D}_t)]_{j=1}^m$) on all tasks across different pre-trained models. For each mask, the amount of selected parameters (10%) is chosen based on the experiment of Figure 1.

| Method | ViT-B/32 | | T5-Small | |
|---|---|---|---|---|
| | Abs. (↑) | Norm. (↑) | Abs. (↑) | Norm. (↑) |
| Pre-trained (Zero-shot) | 47.72 | - | 55.70 | - |
| Non-linear FT (Ilharco et al., 2023) | 71.25 | 76.94 | 65.04 | 87.98 |
| TIES-Merging (Yadav et al., 2023) | 74.79 | 82.84 | 62.53 | 94.83 |
| Task-wise AdaMerging (Yang et al., 2024) | 73.39 | 79.02 | 66.19 | 89.86 |
| Layer-wise AdaMerging (Yang et al., 2024) | 77.06 | 82.98 | 66.61 | 89.86 |
| **TaLoS (Ours)** | 79.67 | 90.73 | 65.04 | 97.22 |
| **TaLoS + TIES-Merging** | 78.15 | 89.10 | 54.54 | 85.42 |
| **TaLoS + Task-wise AdaMerging** | 79.73 | 90.84 | 66.47 | 99.21 |
| **TaLoS + Layer-wise AdaMerging** | **80.25** | **91.40** | **66.76** | **99.63** |

Table 5: **TaLoS on different model merging schemes.** Average absolute accuracies (%) and normalized accuracies (%) of CLIP ViT-B/32 and T5-Small pre-trained models edited by adding task vectors on each of the downstream tasks. We normalize performance of each method by their single-task accuracy. **Bold** indicates the best results. Underline the second best.

between tasks. The number of parameters selected by each mask is 10%, in line with the experiment of Figure 1. Smaller vision models (ViT-B/32) exhibit high parameter sharing ($> 0.7$ mIoU) of low-sensitivity parameters, while smaller language models (T5-Small) share fewer (0.3–0.5 mIoU). However, with a fixed 10% mask sparsity, larger models in both vision and language domains share more low-sensitivity parameters across tasks.

## A.8 COMBINING TaLoS WITH OTHER MODEL MERGING SCHEMES

We extend Table 1 in Table 5 by testing our TaLoS in combination with other merging schemes (TIES-Merging Yadav et al. (2023) and AdaMerging Yang et al. (2024)). Specifically, for TIES-Merging we skip the sparsification part, as the task vectors obtained by TaLoS are already sparse. Regarding AdaMerging, we test both Task-wise AdaMerging and Layer-wise AdaMerging. As we can see, in both vision and language experiments, applying TIES-Merging to our TaLoS is harmful. Seemingly, the signs of task vectors obtained via TaLoS play an important role and disrupting them according to some heuristics causes a drop in performance. Regarding AdaMerging, we can see that TaLoS has full compatibility with existing methods for automating the selection of optimal merging coefficients, highlighting its versatility. However, by itself TaLoS is already robust enough that it doesn't benefit this much from neither task-wise tunings nor layer-wise tunings.

