# OpenReview forum: "Efficient Model Editing with Task-Localized Sparse Fine-tuning"
_ICLR.cc/2025/Conference — ICLR 2025 Poster_

### Official Review · Reviewer_YVQa · 2024-10-29

**Soundness:** 3
**Presentation:** 2
**Contribution:** 3
**Rating:** 6
**Confidence:** 4

**Summary:**

This paper introduces Task-Localized Sparse Fine-Tuning (TaLoS), a novel approach to efficient model editing through sparse fine-tuning, aimed at enhancing task-specific performance in pre-trained models without explicit linearization. The method selectively updates parameters with minimal cross-task impact, achieving weight disentanglement and enabling efficient task arithmetic. Extensive experiments across vision and language domains demonstrate TaLoS's superiority over baseline methods in task addition/negation performance, computational cost, and memory efficiency.

**Strengths:**

+ The paper is well-written, with clear explanations of the motivation.
+ The empirical analyses offer valuable insights into the model’s behavior, which is interesting and well-reasoned.
+ The experiments are comprehensive.

**Weaknesses:**

- Could the authors clarify further how TaLoS achieves linear behavior? The paper mentions that parameters unimportant for a given task are also generally unimportant for other tasks, which might imply a shared subset of parameters across tasks. It would be helpful if the authors provided experimental results showing parameter overlap across task vectors.
- It is unclear how the mask sparsity ratio is determined in practice. A sensitivity analysis of this hyperparameter might be valuable.
- I am curious about how this method might perform in combination with other model merging techniques, such as Ties-Merging or AdaMerging. Although such an investigation would require additional work, discussing this potential integration would be interesting.

**Questions:**

see weekness.

---

> ### Author Response · Authors · 2024-11-24
>
> We thank the Reviewer for their constructive feedback on our work. We appreciate the positive assessment, particularly regarding the thorough testing of our claims, the comprehensiveness of our analysis, and the quality of the writing. We respond below to the weaknesses and questions raised in their Review.
>
> - **(W1) Linear behavior + Parameter overlap across tasks.**
> This question has two parts. Regarding the first part, “linearized behavior” refers to the fact that the output of the model $f(x,\theta_0)$ is well approximated by its first-order Taylor approximation centered around $\theta_0$. Equation 9 in the paper formalizes this concept mathematically. We report it here for convenience. As $\||\theta\_t - \theta\_0\||^2 = \||\tau\_t\||^2 \rightarrow 0$,\
> $$ f(x,\theta_0 + \tau_t) = f(x,\theta_0) + \tau_t^\top \nabla_\theta f(x,\theta_0) + \mathcal{O}(\||\tau_t\||^2) $$\
> The quality of the approximation is thus tied to the norm $\||\tau_t\||^2$: the smaller, the better. Updating only a subset of parameters by masking some gradients (sparse fine-tuning, as done in TaLoS) instead of all the parameters (full fine-tuning), will lead to a smaller norm $\||\tau_t\||^2$, provided that the gradient updates are similar in magnitude for the unmasked parameters between the two fine-tuning paradigms. Intuitively, most of the elements of the weight vector $\theta_0$ will remain fixed during fine-tuning, resulting in many entries in the task vector $\tau_t = \theta_t - \theta_0$ to be zero. Hence, the smaller the norm, the more accurate the approximation. We added a clarification on this aspect at L270.\
> Furthermore, by following [1] we provided empirical evidence of this fact, as discussed at LL1132-1148. In Figure 6 of Appendix A.4 each red marker’s y-coordinate is the single-task fine-tuning accuracy of models obtained by TaLoS while the x-coordinate is the single-task fine-tuning accuracy achieved by linearizing the models obtained by TaLoS. In most cases, adding an explicit linearization to the models obtained by TaLoS doesn’t modify the accuracy, as the markers fall on the bisector (black dashed line). This indicates that the fine-tuning via TaLoS, indeed, exhibited linearized behavior.\
> **In summary, TaLoS achieves linearized behavior by sparsely updating a small subset of parameters that exhibit the lowest sensitivity to a task (measured by $\mathbb{E}\_{x \in \mathcal{D}\_t} [|\nabla\_\theta f(x,\theta\_0)|]$) and keeping fixed the remaining ones.**\
> Regarding the second part of the question, we assess in Figure 11 (Appendix A.7) **the parameter overlap of the shared subset of low-sensitivity parameters across task vectors** identified by TaLoS at 90% sparsity. For each parameter, we compute the mean Intersection over Union (mIoU) of masks, between each task pair: starting from pre-trained parameters $\theta_0$, we predict the mask on task $t$ and then check its intersection over union against the mask predicted on task $t’$ (which acts as a ground truth). A mIoU of 1 signals perfect mask overlap between tasks. The number of parameters selected by each mask is 10%, in line with the experiment of Figure 1.
> Smaller vision models (ViT-B/32) exhibit high parameter sharing (>0.7 mIoU) of low-sensitivity parameters, while smaller language models (T5-Small) share fewer of them (0.3–0.5 mIoU). Larger models in both vision and language domains share more low-sensitivity parameters across tasks.

---

> > ### Author Response · Authors · 2024-11-24
> >
> > - **(W2) Determining mask sparsity ratio & bottom-k ablation in TaLoS.**
> > We agree with the Reviewer that how to determine the mask sparsity ratio could have been stated more clearly. We now provide this information explicitly in the main paper at LL260-261, pointing to Appendix A.1 to improve the overall clarity.
> > The hyperparameter selection paragraph (LL880-895) in Appendix A.1 explains that we searched the optimal sparsity level for TaLoS in the range [10%, 99%], selecting the best sparsity according to task negation performance, as it is a cheap proxy to the degree of weight disentanglement [1].\
> > To provide a more detailed overview of the effect of sparsity on TaLoS we introduced section A.5 in the Appendix. Figure 8 shows the task arithmetic performance vs the single-task accuracy (before addition/negation) while varying the sparsity level. Full (non-linear) fine-tuning corresponds to 0% sparsity. Increasing the sparsity improves the task arithmetic performance, while slightly decreasing the average single-task accuracy, as fewer parameters are updated during fine-tuning. Optimal values for absolute accuracy (in task addition) and target accuracy (in task negation) are observed for a sparsity level of 90% across a variety of models. After 90% sparsity there is a slight drop in both task arithmetic and single-task performance, making such sparsity levels not ideal. Intuitively, if the fine-tuning involves too few weights the resulting entries in the task vector ($\tau_t=\theta_t^\star -\theta_0$) will be mostly zero, reducing the ability to perform task arithmetic effectively.\
> > We can conclude that, like other parameter-efficient fine-tuning methods, our approach trades some single-task performance for parameter efficiency. But this trade-off allows also for superior task arithmetic capabilities for TaLoS (Tables 1, 2) while maintaining competitive single-task accuracy, especially for larger models where the performance drop becomes negligible (Figure 9).
> >
> > - **(W3) Combining TaLoS with other merging schemes.**
> > We thank the reviewer for this interesting suggestion. We agree that it could be beneficial to test other schemes for selecting optimal merging coefficients jointly with TaLoS, given its ability of building high quality task vectors. We choose to test both TIES-Merging and AdaMerging, as suggested by the reviewer.\
> > Specifically, for TIES-Merging we skip the sparsification part, as the task vectors obtained by TaLoS are already sparse. Regarding AdaMerging, we test both Task-wise AdaMerging and Layer-wise AdaMerging.
> > As we can see, in both vision and language experiments, applying TIES-Merging to our TaLoS is harmful. Seemingly, the signs of task vectors obtained via TaLoS play an important role and disrupting them according to some heuristics causes a drop in performance. Regarding AdaMerging, we can see that TaLoS has full compatibility with existing methods for automating the selection of optimal merging coefficients, highlighting its versatility. However, TaLoS is inherently robust, showing minimal benefit from additional task-wise or layer-wise tunings.
> >
> > |                               |   |    **ViT-B/32**   |                     |   |    **T5-Small**   |                     |
> > |-------------------------------|---|:-----------------:|:-------------------:|---|:-----------------:|:-------------------:|
> > |          **Methods**          |   | **Absolute Acc.** | **Normalized Acc.** |   | **Absolute Acc.** | **Normalized Acc.** |
> > | Pre-trained (Zero-shot)       |   |       47.72       |          -          |   |       55.70       |          -          |
> > | Non-linear FT                 |   |       71.25       |        76.94        |   |       65.04       |        87.98        |
> > | TIES-Merging                  |   |       74.79       |        82.84        |   |       62.53       |        94.83        |
> > | Task-wise AdaMerging          |   |       73.39       |        79.02        |   |       66.19       |        89.27        |
> > | Layer-wise AdaMerging         |   |       77.06       |        82.98        |   |       66.61       |        89.86        |
> > | TaLoS (Ours)                  |   |       79.67       |        90.73        |   |       65.04       |        97.22        |
> > | TaLoS + TIES-Merging		          |   |       78.15       |        89.10        |   |       54.54       |        85.42        |
> > | TaLoS + Task-wise AdaMerging	  |   |       79.73       |        90.84        |   |       66.47       |        99.21        |
> > | TaLoS + Layer-wise AdaMerging	 |   |     **80.25**     |      **91.40**      |   |     **66.76**     |      **99.63**      |
> >
> > **References:**
> >
> > [1] Ortiz-Jimenez, Guillermo, Alessandro Favero, and Pascal Frossard. "Task arithmetic in the tangent space: Improved editing of pre-trained models." Advances in Neural Information Processing Systems 36 (2023).

---

> > > ### Comment · Reviewer_YVQa · 2024-11-25
> > >
> > > Thank you for your response. Most of my concerns are addressed.
> > >
> > > I was surprised that TiesMerging approaches AdaMerging with TaLoS on ViT-B/32. AdaMerging is a training-based method, right? Can there be any explanation? Why does this phenomenon disappear when using T5-Small architecture?

---

> ### Author Response · Authors · 2024-11-25
>
> Thank you for the question. Exactly. **Task-wise AdaMerging** seeks to refine task addition by learning a unique merging coefficient, $\alpha_t$, for each task vector $\tau_t$. This approach augments the standard practice, where a single $\alpha$ is determined at validation time for all task vectors [1, 2, 3]. As noted in [2], tuning individual merging coefficients does result in mildly higher performance. However, it comes at the cost of significantly more computation, with relatively modest gains. A variant, **Layer-wise AdaMerging**, further diversifies this by assigning a separate merging coefficient for each layer rather than a global coefficient per task vector. Both AdaMerging approaches introduce additional degrees of freedom, aligning with the logic of TaLoS and generally translating to higher performance.
>
> Regarding **TIES-Merging**, its sign conflict resolution mechanism relies on a heuristic rule applied on $\tau_t$, which may increase the dot product $|\tau\_t^\top \nabla\_\theta f(\cdot,\theta\_0)|$: a quantity that TaLoS seeks to keep minimal, as a dot product closer to zero across $t' \neq t$ tasks indicates that the task vector $\tau_t$ obtained via TaLoS is performing more localized edits primarily in its own task input space with support $\mathcal{D}\_t$ (for more details we point to Equation 6 and discussion at LL182-204).
>
> To investigate empirically whether the sign resolution of TIES is interfering with the logic of TaLoS, we compute $|\tau\_t^\top \nabla\_\theta f(\cdot,\theta\_0)|$ across all tasks $t’=1,..., T$ for both TaLoS and TaLoS + TIES-Merging. For reference, we also report the results for LoTA, with all methods at 90% sparsity. For vision we report ViT-B/32 fine-tuned on RESISC45 and for language we report T5-Small fine-tuned on QASC as no variations in the trend were observed on all the other tasks.
>
> | $\|\tau\_t^\top \nabla\_\theta f(\cdot, \theta\_0)\|$ \| ViT-B/32 (Fine-tuned on RESISC45) \| 90% sparsity |    Cars    |     DTD    |   EuroSAT   |    GTSRB   |    MNIST    |  RESISC45  |   SUN397   |    SVHN    |
> |:------------------------------------------------------------------------------------------------------:|:----------:|:----------:|:-----------:|:----------:|:-----------:|:----------:|:----------:|:----------:|
> | LoTA                                                                                                   |  60243.69  |  46737.64  |   76626.72  |  64460.95  |   54050.29  |  50787.29  |  48023.61  |  42649.41  |
> | TaLoS (Ours)                                                                                           | **611.06** | **335.35** | **1052.58** | **449.79** | **1049.88** | **339.26** | **946.70** | **229.68** |
> | TaLoS (Ours) + TIES-Merging                                                                            |   624.28   |   354.36   |   1097.74   |   487.27   |   1089.99   |   340.76   |   956.02   |   296.87   |
>
> | $\|\tau\_t^\top \nabla\_\theta f(\cdot, \theta\_0)\|$ \| T5-Small (Fine-tuned on QASC) \| 90% sparsity |     QASC    |    WikiQA   |    QuaRTz   |     PAWS    |  StoryCloze |  Winogrande |     WSC    |
> |:--------------------------------------------------------------------------------------------------:|:-----------:|:-----------:|:-----------:|:-----------:|:-----------:|:-----------:|:----------:|
> | LoTA                                                                                               |   1562.98   |   1843.13   |   2438.69   |   1569.33   |   1985.78   |   2240.09   |   905.86   |
> | TaLoS (Ours)                                                                                       | **1365.31** | **1242.91** | **2233.97** | **1267.08** | **1847.66** | **1938.20** | **834.16** |
> | TaLoS (Ours) + TIES-Merging                                                                        |   1384.14   |   1944.48   |   2528.24   |   1585.59   |   2049.51   |   2294.11   |   906.20   |
>
> On ViT-B/32 (RESISC45), TaLoS + TIES-Merging exhibits slightly higher $\tau_t^\top \nabla_\theta f(\cdot,\theta_0)$ values compared to TaLoS, correlating with marginally reduced merging performance (as discussed previously). On language tasks using T5-Small (QASC), the impact of applying TIES-Merging to TaLoS yields significantly higher $\tau_t^\top \nabla_\theta f(\cdot,\theta_0)$, aligning TaLoS + TIES-Merging to LoTA.
>
>
> **References:**
>
> [1] Ortiz-Jimenez, Guillermo, Alessandro Favero, and Pascal Frossard. "Task arithmetic in the tangent space: Improved editing of pre-trained models." Advances in Neural Information Processing Systems 36 (2023).
>
> [2] Ilharco, Gabriel, et al. "Editing models with task arithmetic." International Conference on Learning Representations (2023).
>
> [3] Tang, Anke, et al. "Parameter efficient multi-task model fusion with partial linearization." International Conference on Learning Representations (2024).

---

> ### Author Response · Authors · 2024-12-02
>
> Dear Reviewer,
>
> as the end of the discussion period is approaching, we kindly ask if our responses have adequately addressed their concerns or if there are any additional questions we can clarify.

---

### Official Review · Reviewer_51Ub · 2024-10-31

**Soundness:** 3
**Presentation:** 2
**Contribution:** 3
**Rating:** 5
**Confidence:** 4

**Summary:**

This paper proposes TaLoS, a novel sparse fine-tuning approach for efficient model editing in the task arithmetic framework. The key idea is to prevent interference between tasks by constraining updates to parameters that have low sensitivity across multiple tasks. By selectively fine-tuning only these parameters, the method aims to adapt models to new tasks while minimizing interference when performing task arithmetic operations.

**Strengths:**

- Novel approach combining sparse fine-tuning with task localization for improved task arithmetic

- Comprehensive empirical evaluation across vision and language tasks showing state-of-the-art performance

- Efficiency gains in both computation and memory usage compared to existing methods

- Detailed analysis of weight disentanglement and function localization properties

**Weaknesses:**

1. The caption for Figure 1 lacks clarity, particularly in explaining that "sensitivity" refers to gradient values. This makes the figure difficult to interpret without careful reading of the surrounding text.

2. The key algorithm and mechanism are not presented clearly. For example, there is a grammatical error in the sentence "To prevent interference between tasks and enable task arithmetic, prevents significant changes to the parameters that are highly sensitive to multiple tasks." This verbose phrasing obscures the core mechanism.

3. The paper lacks theoretical analysis or proof of convergence for the proposed method. While empirical results are strong, a theoretical foundation would strengthen the work.

4. Equation 8 introduces parameter k as a significant factor in the upper bound, but there is insufficient discussion of its impact and optimal selection. A more comprehensive analysis of this parameter would be valuable.

5. The direct relationship between gradient values and task interference is not thoroughly explored or justified theoretically. While intuitively reasonable, a more rigorous examination of this connection would bolster the method's foundation.

**Questions:**

- Can the authors provide a clearer, more concise explanation of the core algorithm, perhaps with pseudocode?

- Is it possible to derive theoretical guarantees or convergence properties for the proposed method?

- How sensitive is the method to the choice of k, and are there principled ways to select its optimal value?

- Can the authors elaborate on the direct relationship between gradient magnitudes and task interference, ideally with theoretical justification?

---

> ### Author Response · Authors · 2024-11-27
>
> We thank the reviewer for recognizing the novelty of our work and the comprehensiveness of our empirical validation.
> We address each point raised by the reviewer below:
>
> - **(W1 + W2) Improving the clarity of the paper & general fixes.**
> We updated the caption of Figure 1 to be more clear about the chosen pruning criterion and the sparsity ratio. Regarding the quoted sentence (_"To prevent interference between tasks and enable task arithmetic, prevents significant changes to the parameters that are highly sensitive to multiple tasks."_), **it does not appear in our manuscript, so we are unable to address this concern.** Please see our response to Question 1 (Q1), for clarifications about our approach and the algorithm.
>
> - **(W5 + Q4) Relationship between gradients & task interference.**
> We never claimed in our paper to draw any _“direct relationship between gradient values and task interference”_, so we ask the reviewer to be more specific about this point to be able to address their concern.
>
> - **(W3 + Q2) Theoretical guarantees or convergence properties.**
> Our submission dedicates significant space to the theoretical foundation of our method. We refer the reviewer to Sections 3 and 4 for more details.
> However, our main focus is the empirical evaluation of model merging algorithms for deep learning models (e.g. CLIP and T5) pre-trained and fine-tuned with stochastic gradient descent, which are nonlinear, non-convex settings. Deriving convergence guarantees for these settings is an interesting research direction, but out of the scope of our work.
>
> - **(Q1) Clarifications about TaLoS.**
> Here we re-state for major clarity our TaLoS approach. The pseudocode of TaLoS was provided in Appendix A.2. We made sure to point to it clearly in the main paper (now in L247 of the revised version).\
> First of all, $x$ refers to a single example. Mask calibration consists of estimating which parameters to sparsely update at fine-tuning time. The calibration runs only once per task before fine-tuning the pre-trained model parameters $\theta_0$.\
> As reported at LL239-249 and Algorithm 1, we construct the gradient mask $c \in \mathbb{R}^m$ by setting to 1 the entries in $c$ corresponding to the average output gradient values $s = \mathbb{E}\_{x \in \mathcal{D}\_t} [|\nabla\_\theta f(x,\theta\_0)|] \in \mathbb{R}^m$ that are below the threshold $s_k$. The remaining entries in $c$ are set to 0.\
> The threshold is selected as the $k$-th largest entry in the vector $s$, where $k$ is a user-defined hyperparameter and controls the amount of parameters that will be updated during sparse fine-tuning.
> $s$ can be obtained efficiently using only a small subset of data of the task by computing per-example gradients.\
> The data used in this process is the validation split of the considered task. As detailed in Appendix A.1 at LL894-899, this choice aligns with the setting adopted by LoTA for mask calibration, ensuring a fair and direct comparison between our calibration method and theirs.\
> We remark that the focus is on fine-tuning pre-trained models to produce high-quality task vectors $\tau_t$ that effectively capture the knowledge of every $t$-th downstream task. We can access one task at a time when fine-tuning i.e., we are constrained to individual task training. The data from both the training and validation splits is available in our setting as prescribed by common task arithmetic practices [1, 2].\
> As explained in Appendix A.2 (LL1016-1025, and now also at LL245-248 of the main submission), the mask $c$ could be calibrated by performing a single pass on the data split used i.e., in a single round. However, noisy gradients may affect this process, as evidenced in the Pruning-at-Initialization (PaI) literature [3]. A general solution adopted there consists of running multiple rounds and keeping only the bottom-$p$ parameters (where $p \geq k$) at each round, excluding the rest from mask calibration. Thus, we followed that strategy with an exponential decay of the current sparsity $p$. We repeat this process until we reach the user-defined target sparsity $k$, obtaining the final mask $c$.\
> We hope the algorithm of our approach is clearer now. Should you have any more specific concerns, please let us know and we will be happy to answer. **We also encourage the reviewer to read the responses to other reviewers for further clarification.**

---

> > ### Author Response · Authors · 2024-11-27
> >
> > - **(W4 + Q3) Bottom-k ablation in TaLoS.**
> > In Appendix A.1 (hyperparameter selection paragraph, LL893-895) we explained that the optimal sparsity level for TaLoS was searched in the range [10%, 99%], selecting the best value according to task negation performance as it is a cheap proxy to the degree of weight disentanglement [1]. We are now stating explicitly in the main paper (at LL259-260) that the best results are observed in sparsity ratios between 90% and 99%.\
> > To provide a more detailed overview of the effect of sparsity on TaLoS we introduced section A.5 in the Appendix. Figure 8 shows the task arithmetic performance vs the single-task accuracy (before addition/negation) while varying the sparsity level. Full (non-linear) fine-tuning corresponds to 0% sparsity. Increasing the sparsity improves the task arithmetic performance, while slightly decreasing the average single-task accuracy, as fewer parameters are updated during fine-tuning. Optimal values for absolute accuracy (in task addition) and target accuracy (in task negation) are observed for a sparsity level of 90% across a variety of models. After 90% sparsity there is a slight drop in both task arithmetic and single-task performance, making such sparsity levels not ideal. Intuitively, if the fine-tuning involves too few weights the resulting entries in the task vector ($tau_t=\theta_t^\star -\theta_0$) will be mostly zero, reducing the ability to perform task arithmetic effectively.\
> > We can conclude that, like other parameter-efficient fine-tuning methods, our approach trades some single-task performance for parameter efficiency. But this trade-off allows also for superior task arithmetic capabilities for TaLoS (Tables 1, 2) while maintaining competitive single-task accuracy, especially for larger models where the performance drop becomes negligible (Figure 9).
> >
> > **References:**
> >
> > [1] Ortiz-Jimenez, Guillermo, Alessandro Favero, and Pascal Frossard. "Task arithmetic in the tangent space: Improved editing of pre-trained models." Advances in Neural Information Processing Systems 36 (2023).
> >
> > [2] Ilharco, Gabriel, et al. "Editing models with task arithmetic." International Conference on Learning Representations (2023).
> >
> > [3] Tanaka, Hidenori, et al. "Pruning neural networks without any data by iteratively conserving synaptic flow." Advances in neural information processing systems 33 (2020): 6377-6389.

---

> > > ### Comment · Reviewer_51Ub · 2024-12-01
> > > **Reply by Reviewer**
> > >
> > > Thanks for the reply, I'd like to restate W2: 'a procedure through which we constrain the parameters with the largest ∇θjf(x, θ0) to remain constant and update the only the ones where ∇θjf(x, θ0) ≈ 0.' in line 229.
> > >
> > > Could you provide the results while searching the sparsity level, since it's a significant param in the setting?
> > >
> > > The authors claim that they didn't draw any “direct relationship between gradient values and task interference” while the logic flow of this paper relies on the relationship between gradient values and task interference. Since this relationship has not been widely discussed, the logic is flawed.
> > >
> > > Furthermore, state-of-the-art methods can be broadly categorized into two main types (Yang et al., 2024): (i) Pre-Merging Methods: These methods focus on enhancing the conditions necessary for effective model merging by optimizing the fine-tuning process of individual models. (ii) During Merging Methods: These approaches address task conflicts and interference through various strategies before executing the parameter merging operations. **I agree with Reviewer qd2R that “State-of-the-art task arithmetic solutions are strongly tied to model linearization which leads to computational bottlenecks” is overclaimed.**
> > >
> > > I encourage the authors to discuss the limitations regarding generalization (e.g. other gradient-based optimization) since there's no proof of convergence.
> > >
> > > Yang et al. Model Merging in LLMs, MLLMs, and Beyond: Methods, Theories, Applications and Opportunities.

---

> > > > ### Author Response · Authors · 2024-12-02
> > > >
> > > > > I'd like to restate W2: 'a procedure through which we constrain the parameters with the largest $\nabla\_{\theta\_j} f(x,\theta\_0)$ to remain constant and update the only the ones where $\nabla\_{\theta\_j} f(x,\theta\_0) \approx 0$.' in line 229.
> > > >
> > > > In **W2** the Reviewer asked for clarifications about a sentence not present in our manuscript. Now the revised comment cites a different sentence which we believe is straightforward and does not “obscure the core mechanism.” **Could you kindly clarify the reasoning behind this observation?** Without precise details about the Reviewer’s specific issue we respectfully dispute this concern as it is not possible to provide a comprehensive response.
> > > >
> > > > > Could you provide the results while searching the sparsity level, since it's a significant param in the setting?
> > > >
> > > > We already answered this matter in the previous response (see **(W4 + Q3) Bottom-k ablation in TaLoS**)
> > > >
> > > > > The authors claim that they didn't draw any “direct relationship between gradient values and task interference” while the logic flow of this paper relies on the relationship between gradient values and task interference. Since this relationship has not been widely discussed, the logic is flawed.
> > > >
> > > > We re-state here the logic at the basis of our work. We start from what is observed in [1]: weight disentanglement is an essential condition for task arithmetic and emerges during pre-training. Weight disentanglement implies localization, meaning that updating the model’s weights to learn task t does not affect how the model processes data from other tasks. We propose a sparse fine-tuning strategy that updates only the parameters whose variation does not influence the original functional task localization of the pre-trained model. This leads to effective weight disentanglement and avoids explicit model linearization which was the costly solution proposed in task-arithmetic state-of-the-art literature. This reasoning has been formally presented in Sec. 3 and 4 and widely discussed in Sec. 5 as well as in the Appendix.
> > > >
> > > > We find the statement  ‘direct relationship between gradient values (the Reviewer also used ‘magnitudes’) and task interference’ quite vague and potentially wrong.
> > > >
> > > > Even if we assume the Reviewer was referring to $\nabla\_\theta f(x,\theta\_0)$, the wording of the comment would still be incorrect. In our work, we examine the interaction between **task vectors** and **output gradients** through the dot product $\tau\_t^\top \nabla\_\theta f(x, \theta\_0)$ (Equation 6). This is motivated by the fact that, when a linearized model is presented by input data belonging to task $t$, we want only the corresponding task function (implemented by $\tau\_t^\top \nabla\_\theta f(x, \theta\_0)$) to be active. By fulfilling the constraints of Equation 6 (as TaLoS does), the weight disentanglement property holds, providing guarantees of minimal task interference. Therefore, drawing a direct relationship between gradient magnitudes and task interference would be a significant misunderstanding of the underlying mechanism. The key point is that weight disentanglement, not gradient magnitude comparisons, is the critical factor in mitigating task interference.
> > > >
> > > > We hope the Reviewer can point out which part of the reasoning sounds flawed in their opinion or which equation in the paper appears unclear so that we can provide a targeted explanation.
> > > >
> > > > > Furthermore, state-of-the-art methods can be broadly categorized into two main types (Yang et al., 2024): (i) Pre-Merging Methods: These methods focus on enhancing the conditions necessary for effective model merging by optimizing the fine-tuning process of individual models. (ii) During Merging Methods: These approaches address task conflicts and interference through various strategies before executing the parameter merging operations. I agree with Reviewer qd2R that “State-of-the-art task arithmetic solutions are strongly tied to model linearization which leads to computational bottlenecks” is overclaimed.
> > > >
> > > > We acknowledge the Reviewer’s perspective, but we believe there is a misunderstanding about the aim of our work.
> > > >
> > > > **We focus on task arithmetic that by definition encompasses both addition and negation** [1, 2]. Up to now, only a single work [1] derived theoretical guarantees and empirical proofs of success in task arithmetic by proposing to fine-tune in the tangent space (via explicit linearization). From that work, [3] improved on efficiency by performing partial linearization of the models via custom low-rank adapters.
> > > >
> > > > The Reviewer is referencing a different literature **(model merging), which under the task arithmetic point of view is narrowly focused, addressing only task addition while overlooking task negation** (ref to LL112-115).  We deem it inaccurate to consider model merging solutions dedicated only to task addition as state-of-the-art approaches in task arithmetic.

---

> > > > > ### Author Response · Authors · 2024-12-02
> > > > >
> > > > > > I encourage the authors to discuss the limitations regarding generalization (e.g. other gradient-based optimization) since there's no proof of convergence.
> > > > >
> > > > > We are still not grasping how the point raised by the Reviewer’s comment pertains to the scope of our submission. As already discussed, our work relies on established literature [1, 2, 3] which focus on over-parametrized pre-trained models already converged to a local optimum for all tasks (as mentioned at LL127-130 and reinforced at LL208-210). TaLoS runs a parameter efficient fine-tuning, where the parameters to be updated are chosen in a principled way according to the task arithmetic conditions (weight disentanglement and functional task localization).
> > > > >
> > > > > Can the Reviewer clarify what are the papers presenting ‘other gradient-based optimization’ we should consider as reference and what do they mean with ‘generalization’?
> > > > >
> > > > > **References:**
> > > > >
> > > > > [1] Ortiz-Jimenez, Guillermo, Alessandro Favero, and Pascal Frossard. "Task arithmetic in the tangent space: Improved editing of pre-trained models." Advances in Neural Information Processing Systems 36 (2023).
> > > > >
> > > > > [2] Ilharco, Gabriel, et al. "Editing models with task arithmetic." International Conference on Learning Representations (2023).
> > > > >
> > > > > [3] Tang, Anke, et al. "Parameter efficient multi-task model fusion with partial linearization." International Conference on Learning Representations (2024).

---

> ### Comment · Reviewer_51Ub · 2024-12-02
> **Further questions**
>
> Thanks for the reply. I'd like to re-state my comments point to point.
>
> 1. Grammar errors in line 229, **'...update the only the ones where...'**, which is also the same sentence in my original comment though there's a typo when pasting the sentence.
>
> 2. Thanks for pointing it out, since the sparsity param is very important, I suggest it should be in the main text.
>
> 3. After carefully rereading the paper and rebuttal, I totally agree that task arithmetic comes from weight disentanglement, which could be improved by linear behavior. The authors suggest that sparse fine-tuning leads to linear behavior theoretically whereas I expect some empirical results as the author claimed **'our main focus is the empirical evaluation'.**
>
> 4. I totally respect the proposed method has a broader focus on task negation, however, the authors have mentioned **'model merging'** in sec5 several times, which is a little bit confusing. What's more, it's interesting that authors include Ties-Merging but neglect Ada-Merging which are both during merging methods.
>
> 5. Thanks for restating the foundation of the proposed method, we could conclude that [1,2,3] 'focus on over-parametrized pre-trained models already converged to a local optimum for all tasks'. However, after adding 'regularization' how can we confirm that the model still converges to a local optimum for all tasks? There's a clear performance drop on single-task which is a reflection of robustness. Regarding generalization, I referred to different optimizers as the authors claimed **'modify the update rule of any gradient-based optimization algorithm (e.g. SGD)'.**
>
> Overall, I hope the authors could address all my concerns above.

---

> > ### Author Response · Authors · 2024-12-02
> >
> > 1. We thank the Reviewer for having pointed out a typo, we will make sure to fix it in the final version.
> >
> > 2. The choices regarding the sparsity ratios are already in LL260-261 and point the readers to Appendix A.5. We will make sure to dedicate more space to this information in the main text.
> >
> > 3. At LL273-276 of the main text, we directed readers to Appendix A.4, where we tested the framework proposed in [1] and **confirmed empirically that both TaLoS and LoTA, as sparse fine-tuning methods, achieve stronger linearized behavior than standard non-linear fine-tuning**. This finding is further discussed at LL1132-1148.
> > Figure 6 in Appendix A.4 provides supporting evidence: the red markers plot the single-task fine-tuning accuracies of TaLoS models (y-axis) against the accuracies of the same models after linearization (x-axis). Most markers align with the bisector (black dashed line), indicating that TaLoS inherently achieves linearized behavior, as linearization does not affect its performance. In contrast, LoTA exhibits a weaker degree of linearized behavior, though its markers still fall closer to the bisector than those for standard non-linear fine-tuning.
> >
> > 4. The claim of multiple mentions of "model merging" in Section 5 is incorrect. We **only mentioned “model merging” once, at L308** when presenting LoTA [5] which was primarily proposed for that task. Post-hoc methods like TIES-Merging are fundamentally different from AdaMerging, as they perform heuristic operations on task vectors after fine-tuning to reduce task interference when performing task addition. Instead, AdaMerging is a coefficient selection heuristic for choosing $\alpha_t$ in task addition that replaces standard cross-validation of those coefficients. Hence, it can be applied to all methods in Table 1. However, we agree that it is a meaningful baseline so **we reported experimental results using AdaMerging in Appendix A.8 (Table 5).**
> >
> > 5. Our work strictly follows the methodology of [1,2,3], using the same pre-trained CLIP ViT and T5 models that are assumed to have already converged to a local optimum for all downstream tasks. The pre-training convergence is inherent to the base models, not to our specific intervention. For fairness, we adhered to the AdamW optimizer as prescribed in [1,2,3], ensuring comparability with the relevant literature. As for generalization, we use sparse fine-tuning [4,5], where the operation of masking gradients is by definition agnostic to optimizer choices.\
> > Like other parameter-efficient fine-tuning methods, our approach trades some single-task performance for parameter efficiency. But this trade-off allows also for superior task arithmetic capabilities for TaLoS (Tables 1, 2) while maintaining competitive single-task accuracy, especially for larger models where the performance drop becomes negligible (Figure 9). We do not see why the drop on single-task should be considered a reflection of robustness.
> >
> > > Furthermore, I kindly remind the authors of the page limit in ICLR, the main text should be within 10 pages while the statement should be after the bibliography per CFP.
> >
> > This suggestion is in contrast with the official ICLR 2025 Author Guidelines (https://iclr.cc/Conferences/2025/AuthorGuide) that explicitly encourage a paragraph-long Reproducibility Statement **at the end of the main text (before references)**. Our current placement aligns with this directive.
> >
> > **References:**
> >
> > [1] Ortiz-Jimenez, Guillermo, Alessandro Favero, and Pascal Frossard. "Task arithmetic in the tangent space: Improved editing of pre-trained models." Advances in Neural Information Processing Systems 36 (2023).
> >
> > [2] Ilharco, Gabriel, et al. "Editing models with task arithmetic." International Conference on Learning Representations (2023).
> >
> > [3] Yadav, Prateek, et al. "Ties-merging: Resolving interference when merging models." Advances in Neural Information Processing Systems 36 (2023).
> >
> > [4] Baohao Liao, Yan Meng, Christof Monz. “Parameter-Efficient Fine-Tuning without Introducing New Latency”, ACL 2023
> >
> > [5] Panda, Ashwinee, et al. "Lottery ticket adaptation: Mitigating destructive interference in llms." arXiv preprint arXiv:2406.16797 (2024).

---

> > > ### Comment · Reviewer_51Ub · 2024-12-03
> > > **Updated**
> > >
> > > Thanks for the detailed reply.
> > >
> > > 3. I acknowledge that the linear behavior results are represented in the appendix; however, this is a significant part of the evidence and should be included in the main text.
> > >
> > > 4. In the caption of Table 5, TaLoS has different schemes of model merging, which also indicates that TaLoS fits in the model merging scheme. Plus, it's very interesting to combine TaLoS with adamerging.
> > >
> > > 5. A good task arithmetic performance comes from both single-task performance and weight disentanglement, I totally respect the effort in improving weight disentanglement. However, the single-task performance is the upper bound which is a clear bottleneck.
> > >
> > > Overall, I have updated my score but there are plenty of drawbacks in its current form. I am open to discussing with AC and other reviewers.

---

> ### Author Response · Authors · 2024-12-03
>
> We thank the Reviewer for the discussion and the reconsideration of their initial recommendation. In the following we hope to clarify these final concerns.
>
>
> > I acknowledge that the linear behavior results are represented in the appendix; however, this is a significant part of the evidence and should be included in the main text.
>
> We will make sure to dedicate in the final version more space to this aspect in the main text.
>
> > In the caption of Table 5, TaLoS has different schemes of model merging, which also indicates that TaLoS fits in the model merging scheme. Plus, it's very interesting to combine TaLoS with adamerging.
>
> Task addition is inherently a model merging scheme, where task vectors are combined using a scaled addition operation. In contrast, TaLoS is entirely agnostic to the choice of model merging scheme (be it Task Addition, TIES-Merging, or AdaMerging). The **sole objective of TaLoS is to produce task vectors that when combined guarantee minimal interference**, leaving the specifics of their merging or subsequent use beyond its scope.
>
> > A good task arithmetic performance comes from both single-task performance and weight disentanglement, […] However, the single-task performance is the upper bound which is a clear bottleneck.
>
> As shown in [1], **task arithmetic performance is evaluated across two benchmarks: task addition and task negation** (refer to Tables 1 and 2). While single-task performance can serve as an upper bound for task addition, it does not apply to task negation, where the goal is to reduce the zero-shot performance of pre-trained models. Additionally, Figure 9 highlights that **other task arithmetic methods also trade off minor reductions in single-task performance, sometimes performing worse than TaLoS** (e.g., Linearized FT on ViT-B/32 and Linearized LoRA on T5) for improved task arithmetic performance in both addition and negation. Notably, TaLoS achieves the best performance in these benchmarks (see Tables 1 and 2).
>
> **References:**
>
> [1] Ortiz-Jimenez, Guillermo, Alessandro Favero, and Pascal Frossard. "Task arithmetic in the tangent space: Improved editing of pre-trained models." Advances in Neural Information Processing Systems 36 (2023).

---

### Official Review · Reviewer_S7QQ · 2024-11-02

**Soundness:** 3
**Presentation:** 3
**Contribution:** 3
**Rating:** 6
**Confidence:** 3

**Summary:**

This paper presents a new approach for efficient model editing through Task Arithmetic, specifically using a method called Task-Localized Sparse Fine-Tuning (TaLoS). TaLoS differs from traditional linearization-based approaches by selectively updating only parameters with low gradients, reducing computational load while minimizing task interference. The proposed method enables precise task addition and removal across vision and NLP tasks and demonstrates high performance in experimental results. Additionally, TaLoS achieves a reduced memory footprint and runtime by avoiding explicit linearization, verified by extensive evaluations on computational cost.

**Strengths:**

- Novelty and Effectiveness: This work introduces a novel technique that improves computational efficiency by updating only parameters with low sensitivity while maintaining task separability. This approach significantly reduces the computational burden associated with linearization-based methods and is a valuable contribution in terms of efficiency.
- Empirical Performance: Experimental results demonstrate that TaLoS consistently outperforms baseline methods, including LoTA, across a range of tasks. As shown in Figure 4, TaLoS maintains high accuracy when adding tasks and minimizes the impact on target tasks when removing tasks, showcasing its robustness in task arithmetic.

**Weaknesses:**

- Explanation of Task Interference: Regarding the sensitivity analysis in Figure 1, I understand that it evaluates the performance on other tasks when parameters with low gradients in a given task are pruned. However, could you provide a more detailed explanation on whether altering these selected parameters, rather than simply pruning them, would also have no impact on inference? For example, could you test whether applying extreme random changes to these parameters would still leave other tasks unaffected? This additional analysis would clarify the stability of these parameters across tasks.
- Key Concern: While intuitively, changing parameters with low sensitivity should not significantly affect the function for data drawn from task t , it is less clear why this would not affect other tasks post-finetuning. A detailed explanation of this phenomenon such as measuring the impact of these parameter changes on other tasks' performance or analyzing the overlap of low-sensitivity parameters across different tasks would strengthen the paper.

**Questions:**

- Uniformity of Parameter Sensitivity
Regarding Figure 4, if the gradient norms are not uniform across layers, key parameters might differ across tasks. In this scenario, could the TaLoS approach risk inadvertently pruning important parameters? Could the authors provide further insights on this?

- Conditions for Task-Specific Parameter Pruning
Line 217 mentions that removing parameters deemed unimportant for a specific task does not degrade performance on other tasks. Could the authors clarify the specific conditions or limitations under which this holds? Insights into the effect of model size or architecture variations would also be beneficial.

- Pruning Criteria Unclarity
In Figure 1, the specific pruning criteria (e.g., values for bottom-k) were not readily apparent. Additional detail on this would enhance reproducibility.

---

> ### Author Response · Authors · 2024-11-24
>
> We thank the Reviewer for their constructive feedback. We appreciate the positive assessment, particularly regarding the novelty, efficiency and effectiveness of our method. Below, we address the weaknesses identified by the Reviewer and respond to their questions.
>
> - **(W1) Sensitivity analysis of low-gradient parameters.**
> We are grateful for the interesting question. Applying a perturbation $\theta_0’ \gets \theta_0 + \delta\theta_0$ to a subset of the pre-trained weights in $\theta_0$ and observing no change in the output $f(x, \theta_0’) \approx f(x,\theta_0)$ intuitively means that those weights have low sensitivity to the task. So, pruning or randomizing them would not affect input-output behavior.\
> However, “extreme” randomizations/perturbations would not be suitable to assess the sensitivity of the parameters. If “extreme” randomization refers to very high magnitude perturbations (perhaps, additive), then such perturbations could potentially move the current solution (parametrized by $\theta_0 \in \mathbb{R}^m$) away from the current local optimum, to a distinct region of the loss landscape.\
> Indeed, sensitivity analysis generally refers to “robustness to small perturbations”. This concept, alongside how to perform proper sensitivity analysis on the parameters of a neural network has been formalized by a rich literature dedicated to applications of information geometry [1, 2, 3]. Specifically, as shown by [2, 3], to assess the influence of each weight on the output of a network, we can use the Kullback-Leibler (KL) divergence between the output distribution induced by the original network ($p_{\theta_0}$) vs. the one induced by the perturbed network ($p_{\theta_0 + \delta\theta}$). Mathematically, assuming $\delta\theta \rightarrow 0$ (a small perturbation),\
> $$ D_{KL}(p_{\theta_0} \| p_{\theta_0 + \delta\theta}) = \frac{1}{2} \delta\theta^\top F(\theta_0) \delta\theta + \mathcal{O}(\|\delta\theta\|^3) $$\
> The KL divergence is zero if the perturbation doesn’t affect the output, revealing that the modified weights are not influential for the output. The KL divergence is > 0 otherwise. Here $F(\theta_0) \in \mathbb{R}^{m \times m}$ is the Fisher Information matrix (FIM) [4, 1]. It is a positive semi-definite symmetric matrix defined as,\
> $$ F(\theta\_0) = \mathbb{E}\_x [ \mathbb{E}\_{y \sim p\_{\theta\_0}(y|x)} [\nabla\_\theta \log p\_{\theta\_0}(y|x) \nabla\_\theta \log p\_{\theta\_0}(y|x)^\top] ]  $$\
> It can be used to relate the changes in the parameters to the changes in the outputs. Specifically, it is possible to implement a proper sensitivity analysis of the parameters of a neural network by studying the magnitude of the diagonal elements of the FIM, as they represent the sensitivity of each parameter [2, 3, 5]. Formally, for each parameter $j$, with $j \in 1,...,m$ its corresponding entry on the diagonal of the FIM has value\
> $$ F\_{jj}(\theta\_0) =\mathbb{E}\_x [ \mathbb{E}\_{y \sim p\_{\theta\_0}(y|x)} [\nabla\_{\theta\_j} \log p\_{\theta\_0}(y|x)]^2 ] $$\
> The higher this value is, the more the $j$-th parameter will be sensitive (i.e. will influence the output of the model). Note that in our work we rank parameter sensitivity using $|\nabla_\theta f(x,\theta_0)|$, which is equivalent to using information from the diagonal elements of the FIM. Modeling $p_{\theta_0}$ as a categorical distribution induced by the outputs of $f$ (which is a proper assumption for multi-class classification tasks), the logarithm function ($y = log(x)$), the squaring function ($y = x^2$) and the absolute value function ($y = |x|$) are monotonically increasing in the open interval $]0,+\infty[$. Hence, the sensitivity ranking is the same as $ [\nabla_\theta \log p_{\theta_0}(y|x)]^2$.\
> This said, we repeat in Figure 10 (Appendix A.7) the experiment of Figure 1, but by adding noise distributed as $\mathcal{N}(0, 2\sigma I)$ to the bottom-10% of parameters, instead of pruning them. $\sigma$ is the standard deviation of the parameters, previous to perturbation. We report it in the figure together with the average magnitude $\mu$ prior to applying noise.
> The results align with the analysis reported in Figure 1, highlighting the stability of these parameters across tasks.\
> Moreover, we decided to provide further evidence about the overlap of low-sensitivity parameters across tasks in Figure 11 (Appendix A.7). For each parameter, we compute the mean Intersection over Union (mIoU) of masks, between each task pair. Starting from pre-trained parameters $\theta_0$, we predict the mask on task $t$ and then check its intersection over union against the mask predicted on task $t’$ (which acts as a ground truth). A mIoU of 1 signals perfect mask overlap between tasks. The number of parameters selected by each mask is 10%, in line with the experiment of Figure 1.

---

> > ### Author Response · Authors · 2024-11-24
> >
> > - **(W1) Sensitivity analysis of low-gradient parameters (cont.).**
> > Smaller vision models (ViT-B/32) exhibit high overlap (>0.7 mIoU) of low-sensitivity parameters, while smaller language models (T5-Small) share fewer parameters (0.3–0.5 mIoU). When passing to larger models, in both vision and language domains the overlap of low sensitivity parameters across tasks grows.
> >
> > - **(W2) Interaction of low-sensitivity parameters across tasks.**
> > Thank you for the comment! The reviewer’s intuition is correct. Indeed, in the paper we already measured the impact of these parameter changes on other tasks’ performance. Figure 3 and the discussion in Section 5.2 at LL428-473 confirm that sparsely updating only low-sensitivity parameters with TaLoS doesn’t affect other tasks post fine-tuning (lighter colors signal minimal interference between each fine-tuned model and other tasks’ input spaces).\
> > Intuitively we can explain this because the parameters selected by TaLoS have $\forall x \in \mathcal{D}\_t \nabla\_\theta f(x,\theta\_0) \approx 0$. Indeed, the near-zero gradients suggest a flat region in the loss landscape along the directions corresponding to these parameters. This implies that the model is relatively insensitive to changes within this subspace, regardless of the specific task considered. To provide empirical evidence of our intuition, we computed in the tables below the maximum eigenvalue of the Hessian matrix $\lambda_{\max}$, which is a commonly used metric of local flatness of the loss landscape [8, 9]. A smaller $\lambda_{\max}$ value indicates that the loss landscape is locally flatter.\
> > Specifically, we start from the fine-tuned models on a task $t$ and compute $\lambda_{\max}$ on all tasks $t’=1,...,T$. We consider three mask calibration strategies: TaLoS, LoTA and random mask calibration (“Random”). We use a 90% sparsity ratio for all of them to make the comparison fair. The results confirm our intuition: the $\lambda_{\max}$ values for TaLoS are consistently much lower across all $t' \neq t$ tasks. This indicates that the solution provided by TaLoS is generally more robust to variations, aligning strongly with the findings in Figure 3, where TaLoS retains at least pre-trained performance across all $t' \neq t$ tasks, unlike other methods. Notably, LoTA tends to select parameters that are highly influential for task $t$, but this often correlates with increased sharpness on other tasks ($t' \neq t$), leading to significantly lower robustness, as evident in Figure 3.
> >
> > | Method \| $\lambda\_\max$ on ViT-B/32  \| (Fine-tuned on Cars) |   Cars   |    DTD   |   EuroSAT  |   GTSRB   |   MNIST   | RESISC45 |  SUN397  |    SVHN   |
> > |:-------------------------------------------------------------------:|:--------:|:--------:|:----------:|:---------:|:---------:|:--------:|:--------:|:---------:|
> > | Random – 90% sparsity                                               |   1.40   |   20.82  |   1870.47  |  1357.48  |   137.43  |   13.07  |   28.24  |   26.54   |
> > | LoTA – 90% sparsity                                                 |   0.18   |   77.55  |   709.67   |   248.29  |   168.40  |  106.04  |   47.40  |   166.50  |
> > | TaLoS (Ours) – 90% sparsity                                         | **0.09** | **2.73** | **150.50** | **24.71** | **41.98** | **6.40** | **1.82** | **23.00** |
> >
> > | Method \| $\lambda\_\max$ on ViT-B/32  \| (Fine-tuned on RESISC45) |    Cars   |    DTD   |  EuroSAT  |   GTSRB   |   MNIST   | RESISC45 |  SUN397  |    SVHN   |
> > |:-----------------------------------------------------------------------:|:---------:|:--------:|:---------:|:---------:|:---------:|:--------:|:--------:|:---------:|
> > | Random – 90% sparsity                                                   |   92.94   |   6.83   |   451.23  |   65.31   |    91.7   |   3.99   |   15.58  |   17.05   |
> > | LoTA – 90% sparsity                                                     |   120.12  |   8.00   |   392.46  |   86.72   |   42.31   | **2.78** |   31.44  |   17.35   |
> > | TaLoS (Ours) – 90% sparsity                                             | **88.24** | **1.46** | **64.07** | **27.38** | **37.46** |   2.92   | **0.04** | **16.97** |

---

> ### Author Response · Authors · 2024-11-24
>
> **Answering the questions:**
>
> - **(Q1) Clarifications about the masking strategy.**
> While TaLoS shares similarities with pruning approaches, we remark that it is not a pruning method. Instead, it is a sparse fine-tuning technique that modifies only a subset of the pre-trained model’s parameters by masking their gradients during optimization (e.g., SGD) (LL228-238). Unlike pruning, TaLoS does not remove parameters from the model.\
> The parameters in the subset are those identified as the least important ones for a certain task t with data support $x \in \mathcal{D}\_{t}$, where the concept of “importance” is derived from the output gradient vector $|\nabla\_\theta f(x, \theta\_0)| \in \mathbb{R}^m$ (with the absolute value applied element-wise). For a given data point $x$, the magnitude of the gradient $|\nabla_{\theta_j} f(x, \theta_0)| \in \mathbb{R}$ reflects how strongly the $j$-th parameter influences the output $f(x, \theta_0)$. Larger absolute values indicate greater importance.\
> By selecting parameters with consistently small output gradients, we ensure that only those deemed less critical are updated during fine-tuning, while the more influential parameters remain untouched. **Unlike pruning methods, TaLoS avoids the risk of inadvertently removing key parameters because it preserves all parameters and selectively fine-tunes the unimportant ones.**
>
> - **(Q2 + Q3) Clarifications about the pruning experiment of Figure 1.**
> We updated the caption of Figure 1 to be more clear and specific about the chosen pruning criterion, as well as about the sparsity ratio used which is 10%. This value was found as the maximal sparsity that kept the model’s output almost unchanged, when pruning the least sensitive parameters (sensitivity is measured by $\mathbb{E}\_{x}[|\nabla\_\theta f(x, \theta\_0)|]$ with $x$ drawn from the mask calibration dataset).\
> Our work builds upon the theoretical and empirical foundations established by [6, 7], which position the pre-training regime as a cornerstone of the task arithmetic framework. In particular, [7] established that task arithmetic is a property acquired during pre-training as this learning process not only leads to semantically disentangled feature representations but also to the disentanglement of the weights that govern the output on those semantic sets.
> We considered in our work several model sizes under this specific pre-training regime, on both vision and language domains. We empirically observed a parameter-sharing phenomenon for low-sensitivity parameters across tasks: we can identify them, regardless of the data support (see Figure 1 and the discussion in Section LL213-224).
> This suggests that parameters with low sensitivity to one task tend to exhibit similarly low sensitivity across all other tasks (see also the overlap of such parameters across tasks in Figure 11, Appendix A.7). Notably, model size plays a significant role in this phenomenon: as the model size increases while maintaining a fixed sparsity ratio of 10%, the number of such shared parameters also increases. This is evidenced by lighter regions in Figure 1 for larger models, indicating higher accuracy retention after pruning, and also by the increase in mean Intersection over Union (mIoU) in Figure 11 (Appendix A.7).\
> However, we emphasize that this remains empirical evidence. Accurately characterizing the **precise conditions and limitations of this parameter-sharing phenomenon** for low-sensitivity parameters across tasks, requires further investigation. We leave this as a direction for future work.

---

> > ### Author Response · Authors · 2024-11-24
> >
> > **References:**
> >
> > [1] Amari, Shun-ichi. "Neural learning in structured parameter spaces-natural Riemannian gradient." Advances in neural information processing systems 9 (1996).
> >
> > [2] Chaudhry, Arslan, et al. "Riemannian walk for incremental learning: Understanding forgetting and intransigence." Proceedings of the European conference on computer vision (ECCV). 2018.
> >
> > [3] Pascanu, R. "Revisiting natural gradient for deep networks." arXiv preprint arXiv:1301.3584 (2013).
> >
> > [4] Fisher, Ronald A. "On the mathematical foundations of theoretical statistics." Philosophical transactions of the Royal Society of London. Series A, containing papers of a mathematical or physical character 222.594-604 (1922): 309-368.
> >
> >
> > [5] Matena, Michael S., and Colin A. Raffel. "Merging models with fisher-weighted averaging." Advances in Neural Information Processing Systems 35 (2022): 17703-17716.
> >
> > [6] Ilharco, Gabriel, et al. "Editing models with task arithmetic." International Conference on Learning Representations (2023).
> >
> > [7] Ortiz-Jimenez, Guillermo, Alessandro Favero, and Pascal Frossard. "Task arithmetic in the tangent space: Improved editing of pre-trained models." Advances in Neural Information Processing Systems 36 (2023).
> >
> > [8] Kaur, Simran, Jeremy Cohen, and Zachary Chase Lipton. "On the maximum hessian eigenvalue and generalization." Proceedings on. PMLR, 2023.
> >
> > [9] Chaudhari, Pratik, et al. "Entropy-sgd: Biasing gradient descent into wide valleys." Journal of Statistical Mechanics: Theory and Experiment 2019.12 (2019): 124018.

---

> ### Comment · Reviewer_S7QQ · 2024-11-26
> **Official Comment by Reviewer S7QQ**
>
> Thank you for your detailed response and the additional experiments.
>
> I understand that low-sensitivity parameters have minimal impact and do not interfere with other tasks. However, I am finding it difficult to intuitively understand how fine-tuning such parameters, which have only a small individual influence, can lead to improved performance on specific tasks.
>
> Could you please provide a __concise explanation__ for this? Additionally, could you indicate which figures in the paper best demonstrate this effect?
>
> Addressing these questions would clarify my understanding and make me confident to change my score.

---

> ### Author Response · Authors · 2024-11-28
>
> We thank the Reviewer for the observation. We clarify that the parameters identified by TaLoS are **not strictly null** in influence but rather are **comparatively smaller** in gradient magnitude to others. As noted in Section 4, we rank parameters based on $|\nabla\_{\theta\_j} f(x, \theta\_0)|$ values and select those with smaller magnitudes, where “smaller” is a relative measure, not an absolute one. Thus, affirming $\nabla\_{\theta\_j} f(x, \theta\_0) \approx 0$ does not imply $\nabla\_{\theta\_j} f(x, \theta\_0) = 0$, reinforcing that even parameters with marginal gradients can **collectively** contribute to task performance.
>
> This aligns with prior research in parameter-efficient fine-tuning, which has shown that small parameter updates, as well as the involvement of only a limited subset of parameters, can effectively guide learning. For instance, in [1, 2] transfer learning is achieved by fine-tuning only the bias terms of large pre-trained models. In [3] only pre-trained parameters with the smallest absolute magnitude are trainable.
>
> **References:**
>
> [1] Zaken, Elad Ben, Shauli Ravfogel, and Yoav Goldberg. "Bitfit: Simple parameter-efficient fine-tuning for transformer-based masked language-models." arXiv preprint arXiv:2106.10199 (2021).
>
> [2] Xu, Shichao, et al. "One for many: Transfer learning for building hvac control." Proceedings of the 7th ACM international conference on systems for energy-efficient buildings, cities, and transportation. 2020.
>
> [3] Baohao Liao, Yan Meng, Christof Monz. “Parameter-Efficient Fine-Tuning without Introducing New Latency”, ACL 2023

---

> ### Author Response · Authors · 2024-12-02
>
> Dear Reviewer,
>
> as the end of the discussion period is approaching, we kindly ask if our responses have adequately addressed their concerns or if there are any additional questions we can clarify.

---

### Official Review · Reviewer_qd2R · 2024-11-03

**Soundness:** 2
**Presentation:** 1
**Contribution:** 2
**Rating:** 5
**Confidence:** 4

**Summary:**

The paper proposes a novel fine-tuning strategy based on sparsity that is designed explicitly to facilitate merging a posteriori. The proposed method first identifies a per-task parameter mask that actively promotes weight disentanglement and is used during fine-tuning to constraint the parameter updates only to those in the mask. The method includes vision and NLP benchmarks for task addition and negation as well as weight disentanglement plots which are common in the literature

**Strengths:**

1. The motivation of the paper is clear and the writing is easy to follow
2. The method is simple and intuitive.
3. The performance is superior compared to the closest baselines in both task addition and negation, highlighting the importance of sparsity for task localization and the effectiveness of the method.
4. The experiments on mask calibration are interesting, e.g. Figure 4. and showcase that the method clearly identifies a pattern in the weights of the model.

**Weaknesses:**

The paper has several flaws that, if addressed, can improve the paper significantly. Specifically:


1. The applicability of the method is undermined by the fact that the user needs to redo the fine-tuning from scratch.
2. Mask construction part of the algorithm is not clear: L236 refers to $\mathbf{x}$ but it is not clear if the process involves a single batch or not. The reader needs to go to Algorithm 1 of the appendix to find that multiple “rounds” are used. Both the algorithm and the term “rounds” are never mentioned in the main text. Thus, the reader cannot understand the computational overhead of the method and the assumptions on task availability.
3. The level of mask sparsity is not mentioned on the paper, while it is a major part of the algorithm, apart from the caption of figure 4. It is not clear if 90% is used throughout. Constraining the fine-tuning process too much will result in worse single-task performance (before merging), limiting the method’s applicability. This is an important hyperparameter of the method and it is not discussed/ablated. Moreover, the absolute fine-tuned results should also be given. This is especially important given that fine-tuning in the tangent space “may cause single task performance drop” (L118). The reader can only imply the difference in performance from Figure 3, where the proposed method achieves lower performance compared to some tasks such as SVHN, Cars and DTD.
4. Logical jump in motivation. The paper includes a nice experiment for Figure 1, discussed in lines 208-226. However, this experiment focuses on the *zeroshot* and there is a logical jump on how and why this translates to the fine-tuned checkpoints. The paper would benefit from better explanation
5. Presentation of results is sometimes sloppy. For instance, T5-large Absolute: 77.31 > 76.20, making the +2.87 in the last row wrong.
6. Unfair comparison for task negation: all posthoc methods are designed for task addition, making the comparison unfair imo. this should be stated clearly.
7. Need rephrasing:
	1. “This inherent property of sparse finetuning increases the likelihood that the linearization condition will hold, effectively rendering explicit network linearization unnecessary.” This is not shown and should be framed as intuition rather than fact
	2. Contribution 1 needs to be rewritten and be more specific: the current version can be said for any model merging method
8. Related work: apart from Fisher merging, the methods outlined in the model merging paragraph are proposed primarily for task addition and, therefore, should be merged with the task arithmetic paragraph. The authors should include a paragraph specific to fine-tuning specific for merging, as these are their baselines and most related works.

“Existing task arithmetic solutions are strongly tied to model linearization which leads to computational bottlenecks” → does this refer to tangent? task arithmetic/ties etc have no bottleneck

Minor comments:


1. L47: interference missing refs from ties and tall masks
2. L52: “is crucial for preventing interference” this does not seem like a correct characterization
3. L137-138: this is taken from [1], reference is missing. Moreover, equations 2 and 3 seem redundant
4. L194: fix notation of the subscript

[1] Ortiz-Jimenez, G., Favero, A. and Frossard, P., 2024. Task arithmetic in the tangent space: Improved editing of pre-trained models. *Advances in Neural Information Processing Systems*, *36*.

**Questions:**

1. Can you explain the discrepancy in the normalized results in t5-small? TA is better at absolute compared to tall masks but much lower in normalized.
2. Table 3: why is non-linear ft forward-backward pass time so much more than the proposed method? Given that it does not have the masking of equation 7, shouldn't it be a lower bound to TaLoS? Also, where does the difference in peak memory usage come from? this would be the case if entire layers are excluded from the mask, saving space in optimizer states, but this is not explicitly mentioned.

---

> ### Author Response · Authors · 2024-11-24
>
> We thank the reviewer for their time to review our work and appreciation for the motivations, the simplicity and the effectiveness of our work. We answer to their questions and weaknesses below.
>
> - **(W1 + W2) Applicability of the method & Clarification on mask construction.**
> We appreciate the Reviewer’s feedback and recognize that the main paper may benefit from additional details to improve clarity, as some critical information may have been relegated to the Appendix. To this end, we added more precise information at LL245-248 and L260 about TaLoS (including the usage of multiple rounds and a pointer to the mask calibration details and the pseudocode).\
> Now, we **clarify that TaLoS does not require an additional fine-tuning step** and re-state in the following our mask calibration approach.\
> First of all, $x$ refers to a single example. We clarified this in the manuscript (LL216, 242, 252, and Algorithm 1).
> **Mask calibration** consists of estimating which parameters to sparsely update at fine-tuning time. **The calibration runs only once per task before fine-tuning the pre-trained model parameters $\theta_0$.**\
> As reported at LL239-249 and Algorithm 1, we construct the gradient mask $c \in \mathbb{R}^m$ by setting to 1 the entries in $c$ corresponding to the average output gradient values $s = \mathbb{E}_{x \in \mathcal{D}_t} [| {\nabla}\_\theta f(x,\theta_0) |] \in \mathbb{R}^m$ that are below the threshold $s\_k$. The remaining entries in $c$ are set to 0.
> The threshold is selected as the $k$-th largest entry in the vector $s$, where $k$ is a user-defined hyperparameter and controls the amount of parameters that will be updated during sparse fine-tuning.
> $s$ can be obtained efficiently using only a small subset of data of the task by computing per-example gradients. By leveraging vectorized implementations, getting $s$ costs around $M$ forward-backward passes, where $M$ is the number of mini-batches.\
> The data used in this process is the validation split of the considered task. As detailed in Appendix A.1 at LL894-899, this choice aligns with the setting adopted by LoTA for mask calibration, ensuring a fair and direct comparison between our calibration method and theirs.\
> We remark that the focus is on fine-tuning pre-trained models to produce high-quality task vectors $\tau_t$ that effectively capture the knowledge of every $t$-th downstream task. We can access one task at a time when fine-tuning i.e., we are constrained to individual task training. **The data from both the training and validation splits is available** in our setting as prescribed by common task arithmetic practices [1, 2].\
> As explained in Appendix A.2 (LL1016-1025, and now also at LL245-248 of the main submission), **the mask $c$ could be calibrated by performing a single pass on the data split used i.e., in a single round.** However, noisy gradients may affect this process, as evidenced in the Pruning-at-Initialization (PaI) literature [3]. A general solution adopted there consists in running multiple rounds and keeping only the bottom-$p$ parameters (where $p \geq k$) at each round, excluding the rest from mask calibration. Thus, we followed that strategy with an exponential decay of the current sparsity $p$. We repeat this process until we reach the user-defined target sparsity $k$, obtaining the final mask $c$. As explained at LL1072-1074, employing around 10 iterations per round, with just 4 rounds (i.e. 40 iterations total) already leads to satisfactory results, as using more rounds/iterations does not yield an increase in the overall performance.\
> **This means that the fine-tuning happens only once (no need to “redo fine-tuning”).**\
> **_Computational cost._** Finally, we provide a detailed comparison of the average memory and time costs of TaLoS vs. other fine-tuning methods in the table below. Specifically, we report the timings in seconds (averaged over the 8 vision tasks) and the peak memory usage in Gibibytes of mask calibration and fine-tuning on a CLIP ViT-L/14. These results confirm that paying a small price before training (mask calibration) allows for substantial gains in terms of fine-tuning costs, which also guarantees better task arithmetic performance.
> By focusing only on the comparison between LoTA and TaLoS we notice that the mask calibration time is approximately the same but the cost in terms of memory is very different. Indeed, the mask calibration of LoTA requires performing a few optimization steps and storing the optimizer states before fine-tuning, while for TaLoS we just need to compute per-example gradients.

---

> ### Author Response · Authors · 2024-11-24
>
> | Methods (on ViT-L/14) | Avg. Calibration Time (s) | Avg. FT Time (s) | Tot. Time (s) | Peak Mem. Calibration (GiB) | Peak Mem. FT (GiB) | Peak Mem. Overall (GiB) | Task AdditionAbs. Acc. | Task AdditionNorm. Acc. | Task Negation Target Acc. | Task Negation Control Acc. |
> |:---------------------:|:-------------------------:|:----------------:|:-------------:|:---------------------------:|:------------------:|:-----------------------:|:----------------------:|:-----------------------:|:-------------------------:|:--------------------------:|
> | Non-linear FT         |             -             |      2479.99     |    2479.99    |              -              |        18.6        |           18.6          |          86.09         |          90.14          |           20.61           |            72.72           |
> | Linearized FT         |             -             |      3311.77     |    3311.77    |              -              |        21.3        |           21.3          |          88.29         |          93.01          |           10.86           |            72.43           |
> | L-LoRA                |             -             |      1053.07     |    1053.07    |              -              |         9.7        |           9.7           |          87.77         |          91.87          |           19.39           |            73.14           |
> | LoTA (90% sparsity)   |         **51.84**         |      2592.40     |    2644.24    |             12.9            |        15.4        |           15.4          |          87.60         |          91.89          |           22.02           |            73.22           |
> | TaLoS (90% sparsity)  |           63.04           |    **656.23**    |   **719.27**  |           **7.8**           |       **7.8**      |         **7.8**         |        **88.40**       |        **95.19**        |         **10.63**         |          **73.55**         |
>
> **_Implementation Details for these results:_** For both LoTA and TaLoS, we used batch size 128 for 40 iterations (in detail, 10 iterations per round for TaLoS, with 4 rounds total) as already specified at LL1072-1074. Moreover, we employed gradient checkpointing (as suggested at L1116) which increases the mask calibration time for both methods by 20% but with a substantial saving in terms of memory. The task arithmetic results are in line with Table 1, with no detrimental effect given by the usage of gradient checkpointing.
>
> - **(W3) Mask sparsity level & Single-task fine-tuning results.**
> We thank the Reviewer for pointing out this aspect. In Appendix A.1 (hyperparameter selection paragraph, LL893-895) we explained that the optimal sparsity level for TaLoS was searched in the range [10%, 99%], selecting the best value according to task negation performance as it is a cheap proxy to the degree of weight disentanglement [1]. We are now stating explicitly in the main paper (at LL259-260) that the best results are observed in sparsity ratios between 90% and 99%.\
> To provide a more detailed overview of the effect of sparsity on TaLoS we introduced section A.5 in the Appendix. Figure 8 shows the task arithmetic performance vs the single-task accuracy (before addition/negation) while varying the sparsity level. Full (non-linear) fine-tuning corresponds to 0% sparsity. Increasing the sparsity improves the task arithmetic performance, while slightly decreasing the average single-task accuracy, as fewer parameters are updated during fine-tuning. Optimal values for absolute accuracy (in task addition) and target accuracy (in task negation) are observed for a sparsity level of 90% across a variety of models. After 90% sparsity there is a slight drop in both task arithmetic and single-task performance, making such sparsity levels not ideal. Intuitively, if the fine-tuning involves too few weights the resulting entries in the task vector ($\tau_t=\theta_t^\star -\theta_0$) will be mostly zero, reducing the ability to perform task arithmetic effectively.\
> We can conclude that, like other parameter-efficient fine-tuning methods, our approach trades some single-task performance for parameter efficiency. But this trade-off allows also for superior task arithmetic capabilities for TaLoS (Tables 1, 2) while maintaining competitive single-task accuracy, especially for larger models where the performance drop becomes negligible (Figure 9).\
> Finally, we discuss the single-task performance of TaLoS before task addition as we agree it is important to show these results. We added section A.6 in the Appendix: Figure 9 presents the accuracies obtained by TaLoS (at 90% sparsity) vs. the other fine-tuning strategies.

---

> > ### Author Response · Authors · 2024-11-24
> >
> > - **(W3) Mask sparsity level & Single-task fine-tuning results (cont.).**
> > In almost all cases TaLoS achieves approximately the same performance of full fine-tuning methods (Non-linear FT and Linearized FT), occasionally improving over Linearized FT (ViT-B/32 on SVHN). This is remarkable, as TaLoS updates only a very small subset of parameters, while full fine-tuning (both linearized and non-linear) updates the whole set of model parameters. Furthermore, compared with parameter-efficient fine-tuning methods, which allows for a truly fair comparison (the parameter count is the same across methods), TaLoS improves almost always with respect to Linearized LoRA and matches the performance of LoTA. We remark that the task arithmetic performance of TaLoS is much higher than that of LoTA (see Tables 1 and 2).
> >
> > - **(W4) Clarifications on the pruning experiment of Figure 1.**
> > We apologize if our motivation experiment may have caused any confusion. Here, we provide a more detailed explanation of the rationale to ensure clarity and demonstrate that the reasoning that guided us to TaLoS is consistent and free of logical gaps.\
> > The constraints in Equation 6 require that, when learning the $t$-th task model $\theta_t$, it holds $\tau_t^\top \nabla\_\theta f(x, \theta_0) = 0$ for any $x \in \mathcal{D}\_{t'}$ with $t' \neq t$. This means that the task vector $\tau_t = \theta_t - \theta_0$ must remain orthogonal to $\nabla\_\theta f(x, \theta_0)$ when $x \in \mathcal{D}\_{t'}$ for all $t' \neq t$.
> > To achieve this, $\theta_t$ can be obtained by modifying only the $j$-th parameters of $\theta_0$ where $\nabla_{\theta_j} f(x, \theta_0) \approx 0$, ensuring that $\tau_{t_j}$ is non-zero only for such parameters. However, implementing this would require sampling from all tasks $t' \neq t$ to determine which parameters satisfy $\nabla_{\theta_j} f(x, \theta_0) \approx 0$. This approach is infeasible under individual training constraints.\
> > The experiment in Figure 1 indicates that, once we have identified **the least-sensitive parameters of the pre-trained model** for one task $t$ (those with $\nabla\_{\theta\_j} f(x, \theta\_0) \approx 0$ for all $x \in \mathcal{D}\_t$), we can simply prune them without any significant effect on the other tasks. Thus, the least-sensitive parameters for task $t$ are similarly the least-sensitive for all other tasks $t’ \neq t$.\
> > This implies that we can identify these parameters using only data from task $t$, adhering to individual training constraints. As noted at LL225-227, this regularity allows us to satisfy the localization constraints in Equation 6 because, regardless of the specific $\tau_t$, updates will only affect parameters that ensure $\tau_t^\top \nabla\_\theta f(x, \theta_0) \approx 0$.
> > **In practice, fine-tuned checkpoints are neither involved in this reasoning, nor required.**\
> > Furthermore, in the paper, we measured the impact of these parameter changes on other tasks’ performance **also after fine-tuning** (see Figure 3 and discussion in Section 5.2 at LL428-473), confirming that for our TaLoS, sparsely updating only low-sensitivity parameters doesn’t affect other tasks post fine-tuning (lighter colors signal minimal interference between each fine-tuned model and other tasks’ input spaces), with respect to what happens for other fine-tuning strategies.
> >
> > - **(W5) Presentation of the results.**
> > We thank the Reviewer for kindly pointing out the mistake in Table 1, we promptly fixed it. Indeed, TALL Masks / Consensus is the second best method on T5-Large. Our improvement is of +1.76, which still makes TaLoS the top-performing method.
> > Following a thorough review, we did not identify any additional errors in the presentation of the results. However, we are open to revising the text and addressing any concerns, provided we receive specific feedback on the details that are considered sloppy and may require improvement.
> >
> > - **(W6) Unfair comparison.**
> > Post-hoc methods considered in our work were designed to be applied to task vectors. They present experiments considering the sum arithmetic operation but in the definition of those methods, there is nothing that prevents them to be applied for subtraction. So, our legitimate curiosity was to check their performance on both task addition and negation. To be crystal clear we have added a sentence in the revised paper at L300 that indicates this choice. We believe it is important to provide the community with a thorough analysis of what the literature offers today. We would greatly appreciate guidance or references to some work that according to the Reviewer might ensure fairer comparisons.

---

> > > ### Author Response · Authors · 2024-11-24
> > >
> > > - **(W7) Rephrasing some claims.**
> > > We thank the Reviewer for their suggestions. We updated the claim in LL273-277 indicating that it is supported by experimental results with reference to Appendix A.4. As discussed at LL1132-1155, in Figure 6 the y-coordinate of every marker is the single-task fine-tuning accuracy of the respective model specified by the marker’s color. The x-coordinate is the single-task fine-tuning accuracy achieved by linearizing the models. Most of the red markers of TaLoS fall on the bisector (black dashed line) which confirms that adding an explicit linearization to the models obtained by TaLoS doesn’t modify the accuracy. **We can conclude that explicit linearization is unnecessary.**\
> > > Furthermore, we addressed the Reviewer’s suggestion about the contributions and rephrased more clearly all of them. Specifically, we underlined that our work is the first to formalize the function localization constraints in Equation 6 and design a (sparse) learning model that satisfies them. The constraints enable exact guarantees of weight disentanglement for linearized networks.
> > >
> > > - **(W8) Related works.**
> > > According to the Reviewer’s suggestions we improved the way in which we present and refer to previous works.
> > > In the related work section we moved TIES-Merging and TALL Masks / Consensus from the “Model Merging” subsection to the paragraph dedicated to post-hoc methods in the “Task Arithmetic” subsection. Additionally, we reworked the paragraph dedicated to fine-tuning strategies for task arithmetic, which are indeed our closest competitors (LL118-121).\
> > > In the abstract, we modified the sentence  _“Existing task arithmetic solutions are strongly tied to model linearization which leads to computational bottlenecks”_ to _“**State-of-the-art** task arithmetic solutions are strongly tied to model linearization which leads to computational bottlenecks”_. Among all the existing task arithmetic approaches, Linearized FT and Linearized LoRA are the one producing top results.
> > >
> > > - **Minor Comments.**
> > > 1) We have extensively cited TIES-Merging and TALL Masks in our work. We have now also added their reference in L47.
> > > 2) In [1] the authors proved that weight disentanglement is a necessary condition for preventing interference, making the term “crucial” mathematically precise in this context.
> > > 3) We have added the reference to [1] as required. Moreover, the apparent redundancy of Eq. (3) and (2) is intentional. The goal is to help the reader understand that the similarity between Eq. (5) and Eq. (3) does not imply a direct match between Eq. (5) and Eq. (2).
> > > 4) We do not see any issue in the subscripts. We did not receive complaints from the other Reviewers so we believe that the notation is clear. We are willing to further improve the presentation if possible, but we ask for more precise feedback about what should be updated.
> > >
> > > **Answering the questions:**
> > >
> > > - **(Q1) Explaining the discrepancy in the normalized results.**
> > > The discrepancy arises from the interplay between absolute accuracy and normalized accuracy metrics, as defined in Equation 11. Normalized accuracy is computed by dividing the absolute accuracy achieved by the merged model on a given task $t$ by the accuracy of the corresponding single-task model before merging. The final normalized accuracy is the average across all tasks $t = 1,... , T$.\
> > > In the case highlighted by the Reviewer, the normalized accuracy of TALL Masks / Consensus is higher than Non-linear FT despite a lower absolute accuracy. This is because TALL Masks stores a sparsified version of task vectors, which leads to a drop in the single-task accuracy pre-addition (the denominator in the normalized accuracy calculation). As a result, even with similar absolute accuracy values, TALL Masks achieves higher normalized accuracy due to a lower single-task reference point.
> > > Non-linear FT, on the other hand, starts from higher absolute values, indicating better initial single-task performance. However, due to higher interference in the merged model, its normalized accuracy decreases, reflecting a performance drop relative to the stronger single-task baselines.\
> > > If we agree that the main goal of model merging / task addition is to create a robust multi-task model that retains or surpasses the performance of single-task models, our method demonstrates superior results. It delivers both the highest absolute and normalized accuracy, demonstrating its ability to minimize interference while maintaining strong performance across tasks.

---

> > > > ### Author Response · Authors · 2024-11-24
> > > >
> > > > - **(Q2) Clarifications on computational costs & memory footprint.**
> > > > Equation 7 shows that the gradients of the loss are masked. No masking is involved in the forward pass, thus its cost is the same for Non-linear FT and TaLoS. **Given the very structured sparsity pattern of the mask calibration (see Figures 4 and 5), in TaLoS we are able to effectively freeze most of the layers/parameters.** Hence, there are significantly less gradients to be computed, which makes the backward pass of TaLoS much faster and less memory intensive than the other methods. **This feature of TaLoS was already mentioned at LL518-523.**
> > > >
> > > > **References:**
> > > >
> > > > [1] Ortiz-Jimenez, Guillermo, Alessandro Favero, and Pascal Frossard. "Task arithmetic in the tangent space: Improved editing of pre-trained models." Advances in Neural Information Processing Systems 36 (2023).
> > > >
> > > > [2] Ilharco, Gabriel, et al. "Editing models with task arithmetic." International Conference on Learning Representations (2023).
> > > >
> > > > [3] Tanaka, Hidenori, et al. "Pruning neural networks without any data by iteratively conserving synaptic flow." Advances in neural information processing systems 33 (2020): 6377-6389.

---

> > > > > ### Comment · Reviewer_qd2R · 2024-11-27
> > > > >
> > > > > Thank you for your detailed responses. I reply to the cases I still have some questions/concerns.
> > > > >
> > > > > **W1+W2**: Thank you for your response and for adding the excerpts in your updated manuscript. By “redoing the fine-tuning from scratch”, I was referring to the fact that TaLoS operates on the fine-tuning level and is not compatible with the thousands of checkpoints available online. This is a limitation regarding flexibility inherent to all fine-tuning alternatives.
> > > > >
> > > > > **W3**: Why is negation performance a reliable proxy? The fact that you use task negation is not mentioned in the main text. Thank you for providing Figure 8 - it is illuminating. It shows, however, that there are cases where the method does not yield any benefits, such as T5. In that case and, in general, providing the **normalized** performance with bold numbers can be misleading, e.g., Table 1 TaLoS has bold numbers on normalized accuracy 5% higher than traditional Task arithmetic without an actual increase in absolute performance. In this case, the method proposes to not use publicly available checkpoints, but do fine-tuning from scratch after ablating the mask sparsity level, which is not flexible.
> > > > >
> > > > > **W8**: I still disagree with the statement that “State-of-the-art task arithmetic solutions are strongly tied to model linearization which leads to computational bottlenecks”. For example, TIES or Ada-Merging among others are not. In general, model merging methods mostly assume standard fine-tuning and not linearized fine-tuning due to the associated costs and lack of flexibility. Overall, I think this statement overclaims.
> > > > >
> > > > > **Q2**: In the end, do you freeze whole layers or e.g. 99% of the parameters of a layer? From an implementation perspective this is crucial; in the latter case the layer still needs to be given to the optimizer unless I am missing something. In any case, if even one parameter of the initial layers is not frozen, the backward pass needs to go through the entire computational graph for the backward (`loss.backward()` in Pytorch) and the only thing that changes is the parameters involved in `optimizer.step()`. The backward is the time-consuming part and for this reason I do not understand why non-linear ft takes ~2500sec and TaLoS takes 656sec given that the number of optimization steps is the same. It would be helpful to explain why this happens.

---

> ### Author Response · Authors · 2024-12-01
>
> **W1+W2.**\
> Thank you for the clarification. We think it is important to consider that knowledge interference reduction in model merging can be performed with two different strategies. Either operating after the single-task training phase using post-hoc techniques, or directly during training. TIES-Merging, TALL Masks, DARE and Breadcrumbs belong to the first family of post-hoc methods and reduce knowledge interference generally following heuristics. Linearized Fine-Tuning, or Parameter-efficient fine-tuning approaches like L-LoRA and LoTA are instances of the second family of methods, as well as TaLoS: they act preventively by modifying the fine-tuning strategy so that interference is less likely to happen during merging. Both types of approaches are valuable and present different advantages/drawbacks: possibly post-hoc methods can be considered more flexible, but fine-tuning approaches when guided by principled parameter selection rules offer more guarantees. Finally, we remark that on the basis of the presented results, TaLoS outperforms all its competitors from any of the described families.
>
> **W3.**\
> **Regarding task negation, we are not sure why the Reviewer refers to reliability.**\
> In our answer we mentioned that task negation performance is a **cheap proxy** to the degree of weight disentanglement [1]. Moreover, **at L287 in the main paper** we pointed the reader to the implementation details in Appending A.1 where we reported our choice regarding this hyperparameter selection strategy (LL880-887).
>
> By measuring the task negation performance we simplify the hyperparameter tuning process. If we have $T$ tasks and want to test $N$ values for a hyperparameter, evaluating task negation requires $T \cdot N$ runs. In case of task addition we should consider all possible combinations among task pairs, requiring $N^T$ runs.
>
> In [1] the authors noted the correlation between normalized accuracy on task addition and target accuracy on task negation. Intuitively, if a task vector is able to achieve better (lower) target task accuracy while retaining high control task accuracy, it indicates that the specialized knowledge captured for that task lies in a subspace with minimal task overlap or redundancy [1, 2, 3]. This enables better preservation of specialized knowledge and avoids destructive interference during task addition. Our findings align with this observation. **This is especially evident in Figure 8** where we obtained the same correlation: the lower the target task accuracy in negation, the higher the normalized accuracy in task addition.
>
> > [Figure 8] shows, however, that there are cases where the method does not yield any benefits, such as T5. In that case and, in general, providing the **normalized** performance with bold numbers can be misleading, e.g., Table 1 TaLoS has bold numbers on normalized accuracy 5% higher than traditional Task arithmetic without an actual increase in absolute performance. In this case, the method proposes to not use publicly available checkpoints, but do fine-tuning from scratch after ablating the mask sparsity level, which is not flexible.
>
> **Regarding the presentation of results, we are surprised by the Reviewer’s statement.**
> Tables 1 and 2 follow the same format as those in [1], where bold font is used to indicate the best results in each column. When two methods achieve identical results, both corresponding values are appropriately highlighted in bold. We do not see why this should be considered misleading.
>
> With respect to the competitors, TaLoS achieves higher normalized accuracy, as it retains more of its single-task pre-addition performance. This indicates that our method generates task vectors with lower interference compared to all other approaches. Notably, only on T5-Small (Table 1), TaLoS performs on par with Non-linear FT in terms of Absolute Accuracy and outperforms it in Normalized Accuracy. **We see no issue of flexibility in this behavior.** A user can readily interpret these results and opt for Non-linear FT (e.g., using pre-trained checkpoints) while acknowledging that task interference will result in low normalized accuracy. When scaling to T5-Base and T5-Large, our method demonstrates superior performance in both absolute and normalized metrics.

---

> ### Author Response · Authors · 2024-12-01
>
> **W8.**\
> We acknowledge the Reviewer’s perspective, but we believe there is a misunderstanding about the aim of our work.\
> **We focus on task arithmetic that by definition encompasses both addition and negation** [1, 2]. Up to now, only a single work [1] derived theoretical guarantees and empirical proofs of success in task arithmetic by proposing to fine-tune in the tangent space (via explicit linearization). From that work, [3] improved on efficiency by performing partial linearization of the models via custom low-rank adapters.\
> **Post-hoc methods that perform model merging have so far been narrowly focused, primarily addressing task addition while overlooking task negation** (ref to LL112-115).  Although these methods have shown empirical success in task addition, our work is the first to evaluate their performance also in task negation, offering a more comprehensive perspective. We deem it inaccurate to consider post-hoc methods as state-of-the-art approaches in task arithmetic.
>
>
> **Q2.**\
> The reviewer is right in claiming that, when calling `loss.backward()`, PyTorch’s automatic differentiation engine goes through the entire computational graph to obtain the gradients for each unfrozen parameter. **This happens through the chain rule that applies to activations, not parameters.** Even if the parameters of a layer are frozen (`requires_grad=False`), PyTorch still needs to compute the gradients of the loss **with respect to the outputs** of that layer. However, it skips entirely the gradient computation **with respect to the frozen parameters.**
>
> We state it more formally by providing an overview on the chain rule functioning.\
> Computing derivatives for parameters of the $i$-th layer by applying the chain rule involves the dot product $\frac{\partial L}{\partial w\_i} = \frac{\partial L}{\partial y\_i} \cdot \frac{\partial y\_i}{\partial w\_i}$. Here L is the final loss function, $w\_i$ is a parameter of the layer, and $y\_i$ is the output of the layer. PyTorch just needs to compute and propagate backwards $\frac{\partial L}{\partial y\_i}$, as the derivative with respect to the parameter $w\_i$ **is not required and will be skipped, hence, saving computation**. In fact, if a previous $i-1$-th layer `requires_grad` on its weight $w\_{i-1}$, to compute the derivative of the loss with respect to $w\_{i-1}$ it will use the relationship:
> $$ \frac{\partial L}{\partial w\_{i-1}} = \frac{\partial L}{\partial y\_i} \cdot \frac{\partial y\_i}{\partial y\_{i-1}} \cdot \frac{\partial y\_{i-1}}{\partial w\_{i-1}} $$
>
> Such computation doesn’t involve the term $\frac{\partial L}{\partial w\_i}$. Moreover, both $\frac{\partial y\_i}{\partial y\_{i-1}}$ and $\frac{\partial y\_{i-1}}{\partial w\_{i-1}}$ are “local” computations, involving just the current layer and the following and typically are obtained on the fly using inputs and outputs of the layers, which are cached during forward pass.
>
> As indicated by Figures 4 and 5, for TaLoS most of the layers have zero trainable parameters, which allows to freeze the full layer (in detail, all the MLPs, LayerNorms, Embedding layers and specifically for Vision Transformers the initial Convolutional layer). In the case of attention layers, only the weights producing Queries and Keys are selected (also the biases of Keys in the case of Vision Transformers) which instead enable partial freezing of these layers, i.e., we `retain_graph` and pass to the optimizer’s `parameters` argument only $W\_q$, $W\_k$ and $b\_k$ in attention layers. So, while the forward pass still involves the computation for all layers (to calculate the final output), the backward pass skips the gradient computation for frozen parameters, which in the TaLoS case account for more than 80% of the total amount of weights (for sparsity ratios between 90% and 99%). The remaining ~20% of parameters have a mask with at least a 1, so they need to `require_grad` as gradients need first to be fully computed for the whole parameter and then those gradients get masked before updating the weights (as in Equation 7).
>
> **References:**
>
> [1] Ortiz-Jimenez, Guillermo, Alessandro Favero, and Pascal Frossard. "Task arithmetic in the tangent space: Improved editing of pre-trained models." Advances in Neural Information Processing Systems 36 (2023).
>
> [2] Ilharco, Gabriel, et al. "Editing models with task arithmetic." International Conference on Learning Representations (2023).
>
> [3] Tang, Anke, et al. "Parameter efficient multi-task model fusion with partial linearization." International Conference on Learning Representations (2024).

---

> > ### Comment · Reviewer_qd2R · 2024-12-02
> >
> > Thank you for the detailed answers. I have increased my score.

---

### Author Response · Authors · 2024-12-04
**Global Note - Outcome of the discussion period**

We thank all Reviewers again for their time and effort in providing their valuable feedback on our work.

We clarify general doubts about the placement in literature of our submission:

- Our paper places itself in the **Task Arithmetic literature**, a framework for scalable and cost-effective model editing. Unlike conventional model merging which aims to only create multi-task models, Task Arithmetic focuses on **deriving task vectors that can edit model behavior**, enhancing capabilities via task addition or suppressing unwanted behaviors through task negation. However, interference between tasks remains a significant barrier, as combining independently fine-tuned models often disrupts performance on previously learned tasks.

- The authors of [1] highlighted the importance of **weight disentanglement**, a condition ensuring tasks modify only their respective activation regions. They demonstrate that pre-training inherently promotes some level of disentanglement but fine-tuning may disrupt this property. Prior solutions relying on explicit model linearization have proven computationally expensive and insufficient, as they fail to ensure task-specific activation.

Thus, in our work we propose a principled way to perform fine-tuning and maintain weight disentanglement for task arithmetic:

- We derive a novel set of **function localization constraints** that provide exact guarantees of weight disentanglement on linearized networks, ensuring task-specific activation.

- We empirically observe that least sensitive parameters in architectures pre-trained on large-scale datasets can be consistently identified across tasks. This insight enables satisfying localization constraints under strict individual training assumptions, **without cross-task communication**.

- We introduce Task-Localized Sparse Fine-Tuning (TaLoS), **a mask selection strategy for sparse fine-tuning** that produces task vectors enabled for Task Arithmetic. TaLoS jointly fulfills the localization constraints and induces a linear regime during fine-tuning, all without the computational overhead of explicit network linearization.

Extensive theoretical and empirical analyses confirm that our proposed approach maintains weight disentanglement, prevents task interference, and ensures compatibility between task vectors. This enables robust and efficient task arithmetic, establishing TaLoS as a practical solution for scalable and targeted model behavior editing.

In response to individual reviewers’ requests, we updated our manuscript as follows:

- **Clarification on mask construction & its computational cost [qd2R, 51Ub, YVQa].**
We added more precise information at LL245-248 and L260 about TaLoS (including the usage of multiple rounds and a pointer to the mask calibration details and pseudocode). We now state explicitly at LL259-260 that the best results are observed in sparsity ratios between 90% and 99%. To provide a more detailed overview of the effect of sparsity on TaLoS we introduced section A.5 in the Appendix and presented the results in Figure 8. We extended Table 3, providing a detailed comparison of the memory and time costs of mask calibration in Appendix A.2 (Table 4 and discussion at LL1117-1123).

- **Single-task fine-tuning results [qd2R]**.
We added section A.6 in the Appendix to show and discuss the single-task performance of TaLoS before task addition: Figure 9 presents the accuracies obtained by TaLoS (at 90% sparsity).

- **Rephrasing some parts & reorganization of the related works**.
We updated the claim in LL273-277 referencing earlier the experimental results of Appendix A.4. We addressed the Reviewer’s suggestion about the contributions and rephrased more clearly all of them. According to the Reviewer’s suggestions, we improved how we present and refer to previous works, moving TIES-Merging and TALL Masks from the “Model Merging” subsection to the paragraph dedicated to post-hoc methods in the “Task Arithmetic” subsection. Additionally, we reworked the paragraph dedicated to fine-tuning strategies for task arithmetic (LL118-121) **[qd2R]**. We improved the caption of Figure 1 to be more clear and specific about the chosen pruning criterion and sparsity ratio **[qd2R, S7QQ, 51Ub]**.

- **Sensitivity analysis & overlap of low-gradient parameters [S7QQ, YVQa]**.
We extended the experiment of Figure 1 by providing further evidence of the stability and overlap of low-gradient parameters across tasks in Appendix A.7.

- **Combining TaLoS with other merging schemes [YVQa].**
We dedicated Appendix A.8 to the results when using other schemes for selecting optimal merging coefficients. We choose to test both TIES-Merging and AdaMerging, as suggested by the Reviewer.

For easier reference, we highlighted new or strongly altered sections in a different color in the revised version of our manuscript.

**References**

[1] Ortiz-Jimenez et al "Task arithmetic in the tangent space: Improved editing of pre-trained models" NeurIPS 2023

---

### Public Comment · ~Leonardo_Iurada1 · 2025-02-10
**Acknowledgements**

We would like to extend our sincere thanks to the Program Committee, the Reviewers, and the Area Chair for their careful evaluation and constructive feedback throughout the review process. We deeply appreciate the time and effort invested in helping us improve our work.

We also wish to address the additional comment provided by the Area Chair regarding the tone of some of our responses. We apologize if any of our replies were perceived as aggressive. We have streamlined our communications to convey our points as clearly and efficiently as possible but our intention was never to be confrontational. We fully recognize the importance of maintaining a respectful and collegial tone and will certainly take this feedback to heart in all our future interactions.

Thank you once again for your invaluable contributions, which have significantly enriched our work.

Sincerely,

The Authors

---

### Meta-Review · Area_Chair_LLzz · 2024-12-22

**Metareview:**

The paper presents  a method for improving task arithmetic by selectively updating a sparse subset of parameters during fine-tuning, thereby achieving weight disentanglement and minimizing task interference. The approach falls in the class of methods that requires a modification of the fine-tuning procedure. The results show a promising comparisons to some post-hoc and full fine-tuning methods obtaining strong performance across a variety of benchmarks. The paper had some issues with clarity and inclusion of important details, most of them addressed in the rebuttal. The authors emphasized that their work focused on task arithmetic thus comparisons to model merging methods were not appropriate, some reviewers disagreed. Indeed these two areas have a blurry border, the authors have now included experiments comparing to AdaMerging and Ties Merging combined with their method and in isolation. A lingering concern was the tradeoff in single task performance. Overall the strengths of the paper exceed the weakness thus acceptance is recommended.

**Additional Comments On Reviewer Discussion:**

The reviewers engaged in detailed discussions regarding the clarity of the paper, theoretical justifications, and empirical results. Concerns raised included the lack of explicit exploration of the gradient-task interference relationship and clarity around the sparsity ratio, lack of comparisons to more recent model merging methods like AdaMerging. The authors addresses most of these points through additional explanations and experiments. A remaining concern of one reviewer  51Ub was comparisons to AdaMerging but this appears to have been included in replies to another reviewer and in the appendix highlighting that the method underperforms TaLos and can be combined with TaLos, thus strengthening the case for the paper.  Notably the authors have taken an aggressive approach in some of their replies to reasonable concerns of reviewers and communications to the AC and are suggested to take a different approach in the future.

---

### Decision · Program_Chairs · 2025-01-22

Accept (Poster)